# Highly loaded bimetallic iron-cobalt catalysts for hydrogen release from ammonia

Shilong Chen [1], Jelena Jelic[2], Denise Rein[3,4], Sharif Najafishirtari [1], Franz-Philipp Schmidt[5], Frank Girgsdies [5], Liqun Kang [3], Aleksandra Wandzilak [3], Anna Rabe[1,4], Dmitry E. Doronkin [2,6], Jihao Wang[1], Klaus Friedel Ortega[1], Serena DeBeer [3], Jan-Dierk Grunwaldt [2,6], Robert Schlögl[3,5], Thomas Lunkenbein [5], Felix Studt [2,6] & Malte Behrens [1,4,7] ✉

Ammonia is a storage molecule for hydrogen, which can be released by catalytic decomposition. Inexpensive iron catalysts suffer from a low activity due to a too strong iron-nitrogen binding energy compared to more active metals such as ruthenium. Here, we show that this limitation can be overcome by combining iron with cobalt resulting in a Fe-Co bimetallic catalyst. Theoretical calculations confirm a lower metal-nitrogen binding energy for the bimetallic catalyst resulting in higher activity. *Operando* spectroscopy reveals that the role of cobalt in the bimetallic catalyst is to suppress the bulk-nitridation of iron and to stabilize this active state. Such catalysts are obtained from $Mg(Fe,Co)_2O_4$ spinel pre-catalysts with variable Fe:Co ratios by facile co-precipitation, calcination and reduction. The resulting Fe-Co/MgO catalysts, characterized by an extraordinary high metal loading reaching 74 wt.%, combine the advantages of a ruthenium-like electronic structure with a bulk catalyst-like microstructure typical for base metal catalysts.

The production of ammonia via the Haber-Bosch process transformed the world as it enabled fertilizers to be produced on an industrial scale[1]. 235 Mtons of ammonia were manufactured in 2021, making it the largest volume production chemical. This production might be further boosted in the near future, as ammonia could help mitigate the climate crisis as a carrier and storage material for renewably produced hydrogen, owing to its high hydrogen content and energy density, as well as convenient infrastructure for transportation and storage[2,3]. In this scenario, hydrogen could be intentionally released from ammonia through its decomposition.

In contrast to ammonia synthesis, its reverse reaction, ammonia decomposition, does not have a comparable large-scale industrial application, but has been employed mostly academically to study the reaction mechanism of ammonia synthesis at ambient pressure for over half a century on catalysts designed for the ammonia synthesis reaction[4]. The most active catalysts for ammonia synthesis are Ru-based ones, but iron catalysts are employed commercially due to their lower prices[5]. Similarly, the best performing catalysts for ammonia decomposition are also Ru-based[6–8]. Although iron has a lower activity[9], the commercial aspect makes it highly attractive, hence, Fe-based catalysts for ammonia decomposition have been extensively studied[10], mainly focusing on nitridation of iron species[11,12], support effects[13], promotional effects of bimetallic alloys[14–21], and other promoter effects[22,23]. Recent studies revealed nitrogen desorption as being

[1]Institute of Inorganic Chemistry, Kiel University, Max-Eyth-Str. 2, 24118 Kiel, Germany. [2]Institute of Catalysis Research and Technology, Karlsruhe Institute of Technology (KIT), Hermann-von-Helmholtz-Platz 1, 76344 Eggenstein-Leopoldshafen, Germany. [3]Max Planck Institute for Chemical Energy Conversion, Stiftstrasse 34-36, 45470 Mülheim an der Ruhr, Germany. [4]Faculty of Chemistry, University of Duisburg-Essen, Universtätsstr. 7, 45141 Essen, Germany. [5]Fritz-Haber-Institut der Max-Planck-Gesellschaft, Department of Inorganic Chemistry, Faradayweg 4-6, 14195 Berlin, Germany. [6]Institute for Chemical Technology and Polymer Chemistry, Karlsruhe Institute of Technology (KIT), Engesserstr. 20, 76131 Karlsruhe, Germany. [7]Kiel Nano, Surface and Interface Science KiNSIS, Kiel University, Christian-Albrechts-Platz 4, 24118 Kiel, Germany. ✉e-mail: mbehrens@ac.uni-kiel.de

the rate-determining step on many transition metal catalysts[24], and explain the superiority of Ru by its moderate nitrogen binding energy[25,26]. In this work, we present a synthesis route for Fe-based catalysts inspired by the highly-loaded Haber-Bosch catalyst, identify nitridation as the reason for its moderate activity and demonstrate how nitridation can be suppressed and a nitrogen binding energy similar to Ru can be reached by alloying with Co.

Fe-based catalyst used in the Haber-Bosch process[27] consists mostly of iron (ca 95%) and only a few percent of irreducible structural promoters such as alumina that help maintain a porous microstructure of the so-called "ammonia-iron" enabling a relatively high Fe surface area despite its low dispersion[28]. In recent studies, we proposed a spinel pre-catalyst approach to synthesize highly-loaded iron catalysts by thermal decomposition of co-precipitated precursors as an alternative to the industrial synthesis of fused catalysts[17,18]. Here, we exploit this recipe further and prepared an Fe catalyst based on a $MgFe_2O_4$ spinel pre-catalyst. Reduction of the spinel leads to a Fe/MgO catalyst with high iron loading of 74%, in which MgO fulfills the role of an intermediate between a structural promoter such as the irreducible oxides in the Haber-Bosch catalyst and a classical support (Fig. 1a). This synthesis approach enables catalyst microstructures that are intermediate between typical supported catalysts with low loading and typical bulk catalysts such as the "ammonia iron" in the Haber-Bosch catalysts (illustration in Fig. 1a). Furthermore, basic supports such as MgO are known to promote ammonia decomposition[29].

## Results

### Highly-loaded iron catalyst: spinel approach

The $MgFe_2O_4$ spinel pre-catalyst was prepared according to an optimized co-precipitation route based on the work of Friedel et al.[17], which comprises the co-precipitation of a layered double hydroxide (LDH) precursor of the type $MgFe^{II}Fe^{III}(OH)_6(CO_3)_{0.5} \cdot n\,H_2O$ followed by its thermal decomposition yielding $MgFe_2O_4$ spinel (Fig. 1b). The synthesis recipes of the spinel pre-catalysts and their basic characterizations are described in detail in the methods section and the Supplementary Information (Supplementary Figs. 1–5 and Supplementary Table 1). The reduction of the spinel pre-catalyst was studied by $H_2$-TPR (Supplementary Fig. 6) and isothermal reduction at 600 °C for 5 h was chosen as activation procedure (see XRD of Fe/MgO in Supplementary Fig. 7). In situ XRD patterns of the activation process were analyzed by Rietveld refinement (Supplementary Figs. 8–9 and Supplementary Table 2). Figure 1c shows that the phase-pure $MgFe_2O_4$ was mostly transformed into iron-magnesium-wüstite of the composition $Mg_{0.48}Fe_{0.52}O$ (according to the lattice parameter obtained by Rietveld refinement, see Supplementary Fig. 8) at 450 °C in accordance with the typical two-step reduction of iron oxides[30]. After reaching 600 °C, the spinel phase completely vanished, while the α-Fe phase started to form and reached ~13%, leaving the rest of magnesiowüstite phase $(Mg_{0.44}Fe_{0.56}O)$ (Fig. 1d). After 5 h at 600 °C, the amount of α-Fe increased from ~13% to ~53%. Meanwhile, the magnesiowüstite phase reached a stable composition of $Mg_{0.73}Fe_{0.27}O$, as determined by the Rietveld refinement. The in situ XRD and TPR

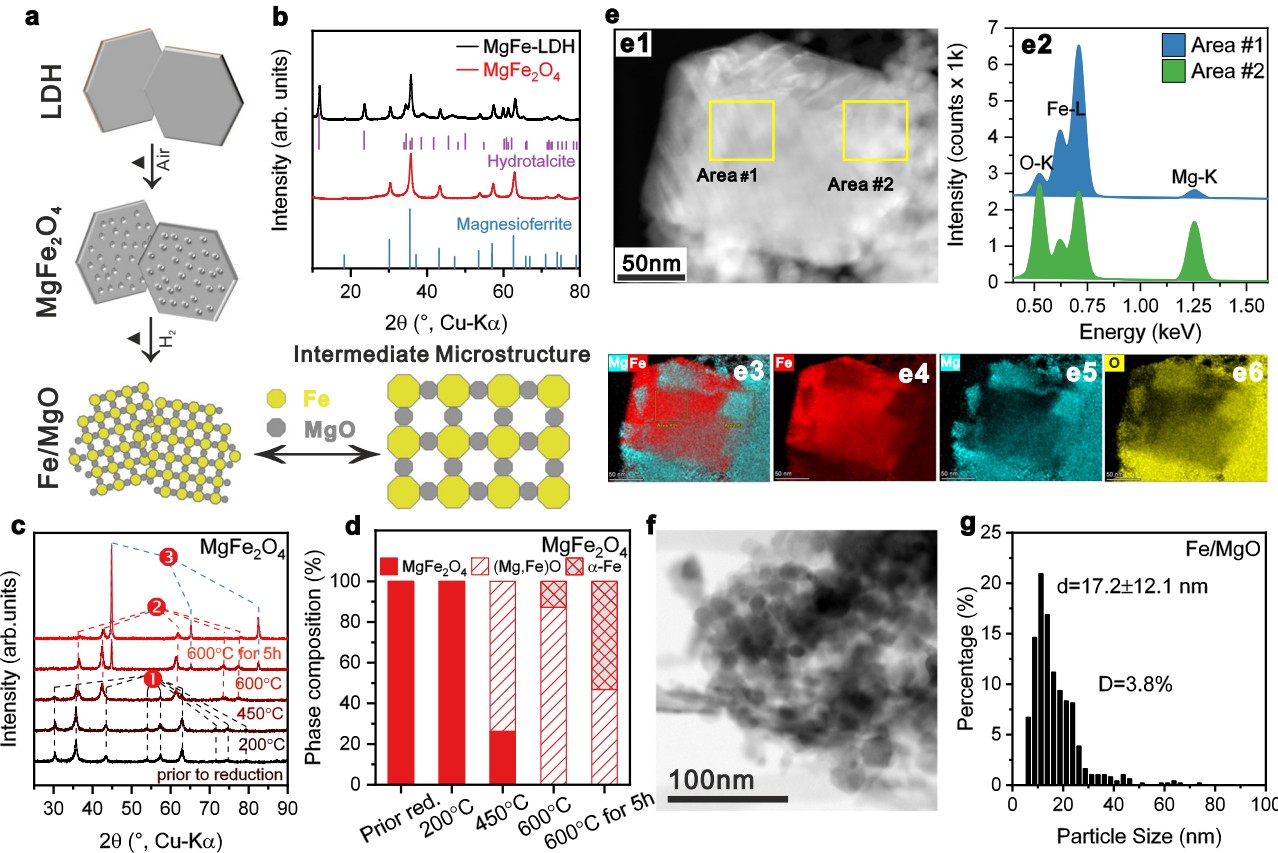

**Fig. 1 | Highly-loaded iron catalyst by spinel approach. a** Scheme of synthesis approach towards an intermediate microstructure between supported and bulk catalyst. **b** XRD patterns of LDH precursor and $MgFe_2O_4$ spinel pre-catalyst. The references: Magnesioferrite (ICSD: 41290), Hydrotalcite (ICSD: 182294) (**c, d**) In situ XRD of the reduction process and the corresponding phase composition transformations during reduction based on Rietveld refinement of the $MgFe_2O_4$, in (**c**) ❶ is $MgFe_2O_4$ phase, ❷ is magnesiowüstite phase, ❸ is α-Fe phase. **e1** Representative HAADF-STEM image of the Fe/MgO catalyst, the corresponding EDS spectra collected at Area #1 and #2 (**e2**), and mapping results with the reconstructed Mg + Fe composition image (**e3**) Fe (**e4**) Mg (**e5**) O (**e6**). The EDS maps are related to K-line intensities from O, Fe and Mg. **f** Representative BF-STEM image of the Fe/MgO catalyst and (**g**) the corresponding metal size distribution, which was determined by the evaluation of at least 400 particles. The error bar represents the standard deviation through particle size statistical analysis. Source data are provided as a Source Data file.

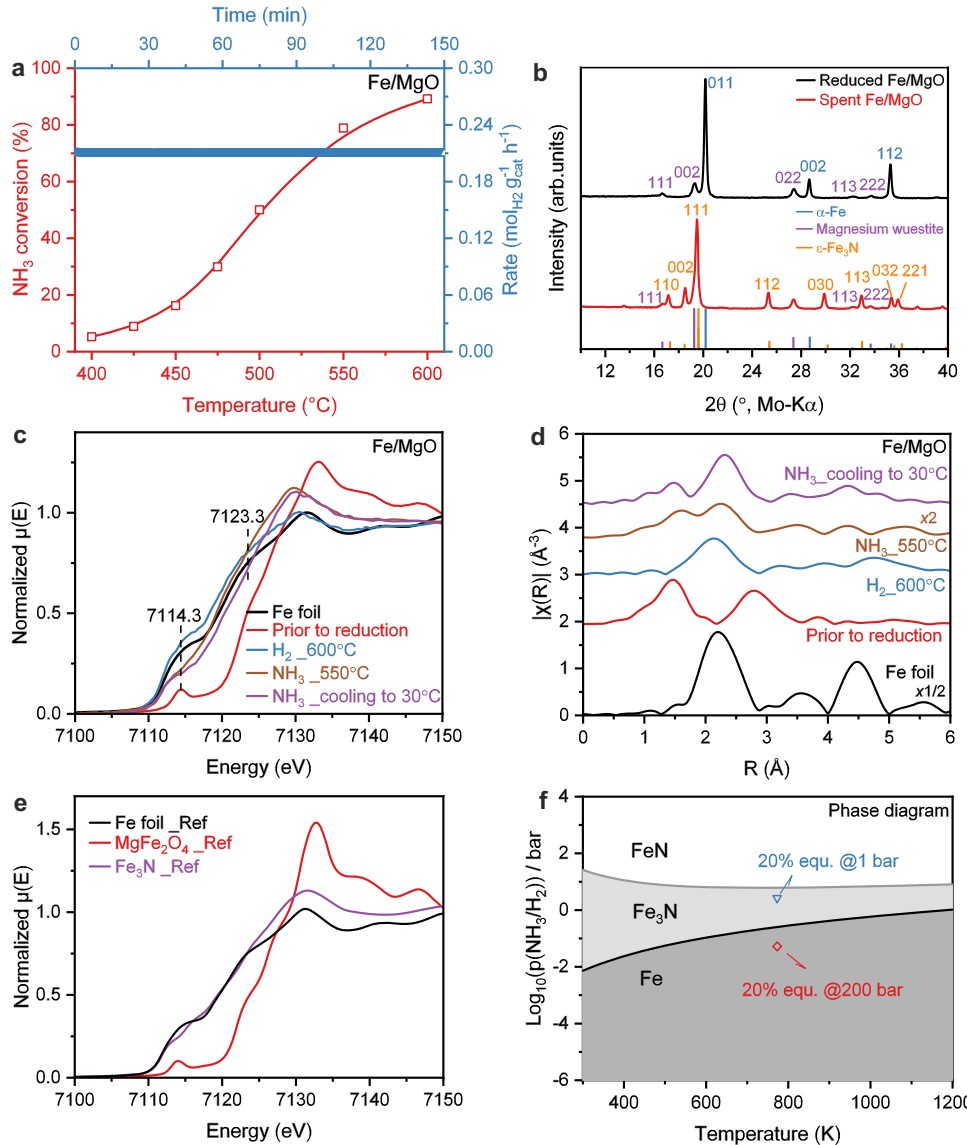

**Fig. 2 | Catalytic activity and structural properties of Fe/MgO catalyst.**
**a** Catalytic activity of Fe/MgO for ammonia decomposition reaction: $NH_3$ conversion (left) and $H_2$ formation rate with time-on-stream (TOS) at 500 °C (right). **b** XRD patterns of the freshly reduced Fe/MgO and the spent Fe/MgO catalyst after reaction condition without exposure to the air. References: α-Fe (ICSD: 52258), Magnesium wüstite (ICSD: 181215), ε-Fe₃N (ICSD: 79981). *Operando* XAS measurement of the Fe/MgO catalyst: (**c**) XANES spectra at the Fe K-edge, (**d**) Fourier transformed $k^2$-weighted EXAFS spectra in R-space at the Fe K-edge. The R-space EXAFS spectra are plotted without phase correction. The corresponding k-space

EXAFS spectra were plotted in Supplementary Fig. 13. **e** Reference Fe K-edge XANES spectra of Fe metal, $MgFe_2O_4$ and $Fe_3N$, which were retrieved from the SPring-8 BL14B2 XAFS Standard Sample Database. The corresponding k-space and R-space EXAFS spectra were plotted in Supplementary Fig. 14. **f** Calculated phase diagram of nitridation of iron. The blue dot represents the condition approaching to 20% of equilibrium for decomposition of $NH_3$ at 500 °C under 1 bar while the red dot represents the condition approaching to 20% of equilibrium for reaction of $N_2$ and $H_2$ ($N_2$:$H_2$ = 1:3) at 500 °C under 200 bar. Source data are provided as a Source Data file.

results thus indicate that a considerable fraction of at least 20% of the total iron content is difficult to reduce and remains in the (Mg,Fe)O solid solution state. Nevertheless, for the sake of brevity, we will denote the support phase as MgO and reduced catalysts as Fe/MgO.

## Highly-loaded iron catalyst: catalytic performance and characterization

The phase composition of the catalyst on the nano-scale has been studied by STEM-EDS. As shown in Fig. 1 e1, a large isolated particle (120×100 nm) was surrounded by a number of smaller particles. Based on the EDS results (Fig. 1 e2-e6), the large particle was assigned to iron metal and the smaller surrounding particles to the oxide phase (Mg,Fe)O. The microstructure of these composite catalysts was further investigated with TEM and the aforementioned

intermediate microstructure between a bulk and a supported catalyst becomes apparent from Fig. 1f and Supplementary Fig. 23a1 showing typical aggregates of larger metallic iron and smaller oxidic support particles. The iron particle size analysis revealed a monomodal, but broad size distribution with an average particle size around 17.2 ± 12.1 nm (Fig. 1g and Supplementary Fig. 23a2). As noted before, opposed to conventional supported catalysts, the oxide particles do not form a typical continuous porous material, but are individually dispersed between the larger metal particles acting as spacers between them, thus rather functioning like a typical structural promoter in bulk catalysts. Such an "intermediate microstructure" of highly loaded base metal catalysts is known for example from the industrial Cu/ZnO catalyst for methanol synthesis as a result of a similar co-precipitation method[31,32].

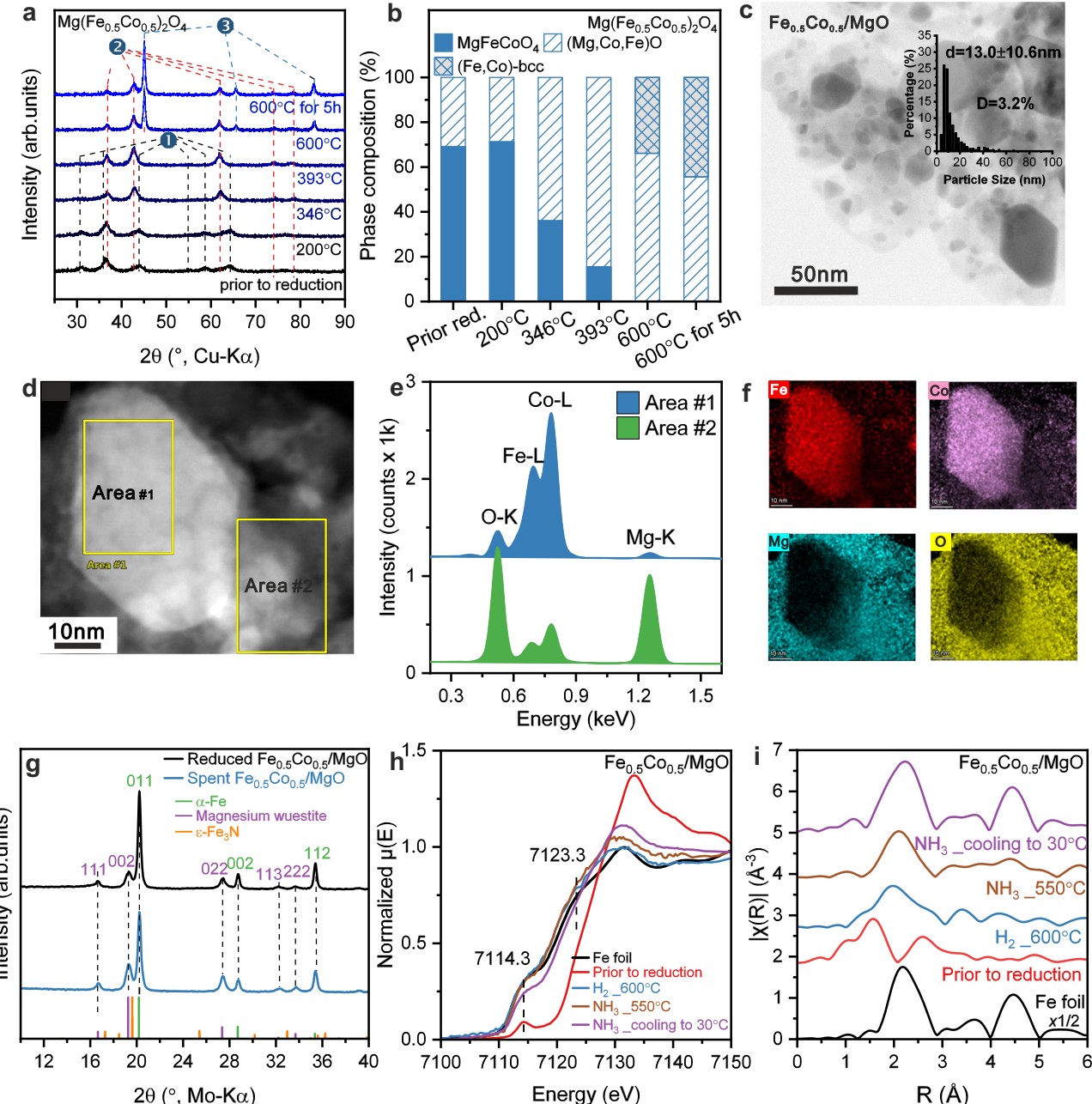

**Fig. 3 | Structural properties of the Fe₀.₅Co₀.₅/MgO catalyst. a** In situ XRD of the reduction process and (**b**) the corresponding phase composition transformations during reduction based on Rietveld refinement of the Mg(Fe₀.₅Co₀.₅)₂O₄, in (**a**) ❶ is MgFeCoO₄ phase, ❷ is Mg(Fe,Co)O wüstite-like phase, ❸ is FeCo-bcc phase. **c** Representative BF-STEM image of the Fe₀.₅Co₀.₅/MgO catalyst and the FeCo alloy particle size distribution, which was determined by the evaluation of at least 400 particles (inset in (**c**)). The error bar represents the standard deviation through particle size statistical analysis. **d** The STEM image and (**f**) the corresponding EDS spectra collected at Area #1 and #2 (**e**) as well as EDX maps, which are related to K-line intensities from O, Fe, Co and Mg. (**g**) XRD patterns of the freshly reduced Fe/MgO and the spent Fe/MgO catalyst after reaction condition without exposure to the air. references: α-Fe (ICSD: 52258), Magnesium wüstite (ICSD: 181215), ε-Fe₃N (ICSD: 79981). *Operando* XAS measurement of Fe₀.₅Co₀.₅/MgO catalyst during reaction: (**h**) XANES spectra at the Fe K-edge, (**i**) Fourier transformed k²-weighted EXAFS spectra in R-space at the Fe K-edge. The R-space EXAFS spectra are plotted without phase correction. The corresponding k-space EXAFS spectra were plotted in Supplementary Fig. 13. Source data are provided as a Source Data file.

The catalytic performance of the Fe/MgO catalyst in diluted ammonia showed stable activity in a temperature window from 400 to 600 °C (Fig. 2a). The steady-state catalyst mass-normalized H₂ production rate at 500 °C reached 0.21 mol$_{H2}$ g$_{cat}^{-1}$ h$^{-1}$ (Fig. 2a), which turned out to be in the moderate to upper range of performance reported previously for Fe-based catalysts (Supplementary Table 5). A reason for this relatively low activity was found in the structural analysis of the spent catalyst with using ex situ XRD and XES and *operando* XAS. The XRD pattern of spent

Fe/MgO catalyst without exposure to the air showed clear reflections of Fe₃N and iron-magnesium-wüstite, with comparison to the freshly reduced Fe/MgO (Fig. 2b). Furthermore the Fe Kβ valence-to-core X-ray emission spectrum (VtC XES) of the spent Fe catalyst is also consistent with the nitridation and supports the transformation of iron to iron nitride after catalysis (Supplementary Fig. 10)[33,34].

To investigate the reason and check for possible reversible phase transformations, *operando* X-ray absorption spectroscopy (XAS) was

employed to study the electronic and geometric structure of Fe and Co species in the catalysts during the reaction. Figure 2c, d shows X-ray absorption near-edge structure (XANES) spectra and Fourier transforms (FTs) of the extended X-ray absorption fine-structure (FT-EXAFS) spectra collected for the Fe/MgO catalyst under reaction conditions at the Fe K-edge. The XANES of $MgFe_2O_4$ showed a sharp pre-edge feature at 7114.3 eV resulting from the 1 s to 3d transitions[35], and a rising edge position at 7123.3 eV (determined by energy position at half absorption of the edge in normalized XANES), which is shifted by -3.8 eV compared to Fe foil. The edge shape and position of the $MgFe_2O_4$ pre-catalyst are nearly identical to those of the commercial spinel $MgFe_2O_4$ reference (Fig. 2e), which has a mixture of tetrahedrally ($T_d$) and octahedrally ($O_h$) coordinated $Fe^{3+}$ sites in the lattice. After reduction in $H_2$ at 600 °C, the XANES edge position of the reduced Fe/MgO catalyst shifted back to a position very close to that of metallic Fe and the EXAFS spectra can be fitted by two Fe-Fe paths derived from the α-Fe structure (at 2.47 ± 0.01 and 2.86 ± 0.01 Å, respectively) without contribution from Fe-O coordination (Supplementary Fig. 16e, f and Supplementary Table 3), suggesting that metallic Fe is the predominant component in the bulk. This shows that the reduction reaches completeness faster in the quartz capillary reactor used for XAS measurement compared to the fixed bed reactor likely due to the much smaller amount of sample. Switching to $NH_3$ atmosphere at 550 °C led to clear changes in the XANES region as compared to the reduced Fe/MgO catalyst, including the decreased intensity of pre-edge feature between 7112 and 7118 eV, disappearance of the shoulder peak at around 7125 eV and a rising white line feature at about 7129 eV (Fig. 2c). In the corresponding EXAFS spectra, a much shorter coordination from a light scattering atom was found, along with a shift in the Fe-Fe shell to a longer distance (Fig. 2d). Based on the EXAFS fitting results, the first coordination shell attributed to a Fe-N/O scattering path with a coordination number (CN) of 1.5 ± 0.2 at 2.04 ± 0.02 Å (Supplementary Fig. 16g, h and Supplementary Table 3), which is shorter than the typical Fe-O bond in both $Fe^{2+}$ and $Fe^{3+}$ oxides, but also longer than the Fe-N distances in the known Fe nitride categories (Supplementary Table 4). The second shell is fit as an Fe-Fe scatterer with a CN of 3.7 ± 0.7 and an average distance of 2.73 ± 0.02 Å, which is very similar to the Fe-Fe distance of 2.71 Å in $Fe_3N$ (Supplementary Fig. 16g, h, Supplementary Table 3–4). The noticeable differences in EXAFS indicate significant structural changes that on a first sight resemble formation of an oxide from the metallic phase. However, this is excluded by the XANES edge position of Fe/MgO under $NH_3$ decomposition condition, which is not consistent with conversion to an oxide phase and lead to the most conceivable conclusion that the changes in electronic structure (from the XANES) and in coordination environment (from the EXAFS) were caused by nitridation of the metallic Fe phase. Such phenomenon could also be supported by comparing the XANES and EXAFS spectra of $Fe_3N$ and Fe foil references (Fig. 2e). In the last condition when Fe/MgO catalyst was cooled down from 550 to 30 °C in $NH_3$, the absorption edge shifted by -1.1 eV to higher energy, while still maintaining most of the edge features and a relatively more prominent pre-edge shoulder peak (Fig. 2c). Despite large differences between the Debye-Waller factors ($\sigma^2$) at 550 and 30 °C, the CNs for both Fe-N/O and Fe-Fe bond were just slightly increased (Supplementary Fig. 16g–j, Supplementary Table 3). However, a discernible reduction in the first shell Fe-N/O distance to 1.95 ± 0.02 Å was observed from the EXAFS fitting results, while the second shell Fe-Fe distance remains constant at 2.72 ± 0.01 Å, which is still similar to the Fe-Fe distance of 2.71 Å in $Fe_3N$ (Supplementary Fig. 16 I, j, Supplementary Table 3). Such structural transformations hint to a transformation of the nitride formed at 550°C to a more stable phase at room temperature.

*Operando* XAS, ex situ XRD, and XES thus were all consistent with the hypothesis that the working state of the monometallic Fe catalysts

in ammonia decomposition is an Fe nitride phase, even in diluted ammonia. Indeed, DFT calculations provide further support for the nitridation of Fe under typical ammonia decomposition conditions (20% approaching to equilibrium at 500 °C, 1 bar) as shown in Fig. 2f and Supplementary Fig. 25. Importantly, this is different from typical ammonia synthesis (e.g. 200 bar, 500 °C, Supplementary Fig. 25), where the bulk-unnitrided "ammonia iron" was reported to be stable[28]. This indicates that also the surface chemistry, including key parameter like the nitrogen binding energy, might be different for iron catalysts used in ammonia decomposition compared to ammonia synthesis. We calculated the nitrogen binding energy of $Fe_3N$ to be −0.37 eV relative to gas-phase $N_2$, which is lower than both the value for metallic Fe (considered too strong) and metallic Ru (considered optimal)[26]. Thus, the challenge in Fe-based ammonia decomposition is either to increase the nitrogen binding energy of Fe nitride or to suppress its nitridation while lowering the nitrogen binding strength of unnitrided Fe at the same time.

## Bimetallic catalyst: alloying iron with cobalt

We have chosen the latter approach and studied the alloying of Fe with a second base metal that shows a lower nitrogen binding energy than Fe and Ru, and at the same time is not forming nitrides easily. Hence, we chose cobalt and employed the above-described synthesis method to synthesize $Mg(Fe_{1-x}Co_x)_2O_4$ spinel pre-catalysts (synthesis recipe: Supplementary Fig. 1, XRD: Supplementary Fig. 2) with different Fe:Co ratios yielding bi-metallic catalysts $Fe_{1-x}Co_x$/MgO (x = 0.25, 0.5) and Co/MgO (x = 1) in addition to the already presented Fe/MgO (x = 0). XRD characterization shows that the reduced cobalt-containing catalysts are comprised of a single metallic phase, which depending on x is α-Fe, bcc Fe-Co alloy, or fcc Co, together with an oxidic wüstite-like Mg(Fe,Co)O phase, which still contains some transition metal cations similar to the pure Fe/MgO (x = 0) catalyst (Supplementary Fig. 7).

The comparative characterization data for the $Fe_{0.5}Co_{0.5}$/MgO catalyst ($Mg(Fe_{0.5}Co_{0.5})_2O_4$ pre-catalyst) is shown in Fig. 3. In situ XRD of the activation behavior was similar (Fig. 3a, Rietveld refinements in Supplementary Fig. 9b and Supplementary Table 2), however, the pre-catalyst was found to be not entirely phase-pure, but it contained wüstite-like Mg(Fe,Co)O already before reduction, whose phase content increased with the activation (Fig. 3b). TEM showed that the $Fe_{0.5}Co_{0.5}$/MgO catalyst features a similar "intermediate microstructure" like the Fe/MgO sample being comprised of larger metal and smaller oxide particles (Fig. 3c). The metal particle sizes show a broad distribution and were found to be slightly smaller than in Fe/MgO catalyst with a size of 13.0 ± 10.6 nm (Fig. 3c inlet). Importantly, the local STEM-EDS measurements (Fig. 3d) and EDS maps (Fig. 3e, f) demonstrate that the big metal particles (area #1) contain both metals Fe and Co, which are homogeneously distributed in one particle. This supports the formation of bimetallic alloy particles. Meanwhile the surrounding particles (area #2) mostly contain Mg and O with a small fraction of Fe and/or Co supporting the formation of wüstite like Mg(Fe,Co)O. The EDS maps and the in situ XRD for activation process of $Fe_{0.75}Co_{0.25}$/MgO proved the similar "intermediate microstructure" as the $Fe_{0.5}Co_{0.5}$/MgO catalyst (see more information in Supplementary Fig. 9a, Supplementary Fig. 11, and Supplementary Table 2).

The XRD pattern of spent $Fe_{0.5}Co_{0.5}$/MgO catalyst after ammonia decomposition reaction showed similar reflections to the freshly reduced $Fe_{0.5}Co_{0.5}$/MgO, suggesting metallic FeCo alloy remained during reaction without nitridation (Fig. 3g). *Operando* XAS studies of $Fe_{0.5}Co_{0.5}$/MgO catalysts were also carried out under the same reaction conditions. The $Mg(Fe_{0.5}Co_{0.5})_2O_4$ pre-catalyst measured at 30 °C exhibited similar XANES and EXAFS spectra to the $MgFe_2O_4$ pre-catalyst. Upon exposing to $H_2$ 600 °C, $Fe_{0.5}Co_{0.5}$/MgO catalyst was almost fully reduced to metallic FeCo, featuring the absorption edge nearly overlapping with that of Fe foil. A similar transformation was observed from the corresponding Co K-edge XANES and EXAFS results

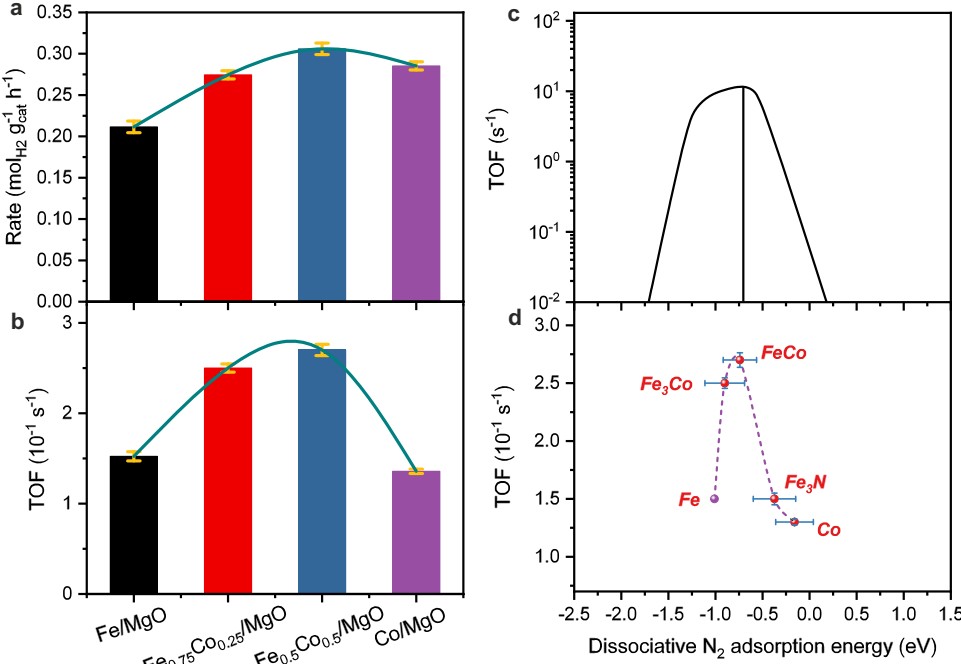

**Fig. 4 | Catalytic performance of various Fe$_{1-x}$Co$_x$/MgO catalysts and their relationship with the corresponding calculated dissociative N$_2$ adsorption energy.** Steady-state reaction rate of H$_2$ production (**a**) and TOF (**b**) of the four catalysts for ammonia decomposition at 500 °C. **c** Calculated TOF of ammonia decomposition at 500 °C, 1 bar, 20% NH$_3$ as a function of the dissociative N$_2$ adsorption energy as reported in the literature[26]. **d** Experimental TOFs of the four Fe$_x$Co$_{1-x}$/MgO catalysts as a function of the corresponding dissociative N$_2$ adsorption energy as obtained from DFT calculations. Note that the Fe pure point is

hypothetical, as our measurements indicate that Fe$_3$N is the active material for this catalyst, purple dashed curve is there to guide eyes. The error bars of reaction rates / TOFs represents standard deviation through two repeated measurements, while the error bars of calculated dissociative N$_2$ adsorption energy represents the calculated standard deviation of the appropriately weighted difference of ensembles provided in the BEEF-vdW functional[49,51]. Source data are provided as a Source Data file.

(Supplementary Fig. 12). Unlike the Fe/MgO catalyst, no noticeable changes of Fe$_{0.5}$Co$_{0.5}$/MgO could be observed in either in the XANES or in the EXAFS spectra during NH$_3$ decomposition reaction at both Fe K-edge and Co K-edge with the comparison to the condition under H$_2$ reduction at 600 °C (Fig. 3h, i and Supplementary Fig. 12), providing experimental evidence that introducing Co to the catalyst significantly enhances the stability of the metallic Fe phase against bulk nitridation. To assign the observed suppression of nitridation to the formation of the alloy, a physical mixture of Fe/MgO and Co/MgO was also tested in ammonia decomposition. The activity of this mixture was between the activity of pure Fe/MgO and Co/MgO (Supplementary Fig. 22a). After ammonia decomposition, the XRD of the spent catalyst mixture showed the disappearance of the α-Fe reflections while crystalline Fe nitrides formed, including Fe$_3$N and Fe$_4$N (Supplementary Fig. 22b). Therefore, the suppression of iron nitridation can be assigned to the formation of the alloy and not to the presence of unalloyed cobalt alone.

## Bimetallic catalyst: relationship between activity and N-binding energy

Comparing the catalytic performance of all four Fe$_{1-x}$Co$_x$/MgO catalysts, the Fe/MgO catalyst showed the lowest ammonia conversion for the reasons outlined above. The substitution of ~50% Fe with Co in Fe$_{0.5}$Co$_{0.5}$/MgO resulted in a remarkable improvement in the activity (Supplementary Figs. 17–18). However, the Co/MgO catalyst with the full substitution of Fe exhibited similar ammonia conversion as the bimetallic catalysts. In order to take the different metal dispersions of the four catalysts into account, the reaction rates of H$_2$ formation at 500°C were carefully evaluated and compared among these four catalysts, first on a catalyst-mass basis (Fig. 4a and Supplementary Fig. 18). The steady-state catalyst mass-normalized H$_2$ production

rate of the Fe/MgO catalyst was clearly increased to ~0.27 and ~0.31 mol$_{H2}$ g$_{cat}^{-1}$ h$^{-1}$ for the two Co substituted samples, Fe$_{0.75}$Co$_{0.25}$/MgO and Fe$_{0.5}$Co$_{0.5}$/MgO, respectively. For the pure Co/MgO catalyst, the rate decreased slightly again to ~0.29 mol$_{H2}$ g$_{cat}^{-1}$ h$^{-1}$ giving rise to a volcano-like evolution of the rates with increasing Co content (Fig. 4a). Furthermore, the Fe$_{0.5}$Co$_{0.5}$/MgO catalyst kept a stable reaction rate at 500 °C (~0.30 mol$_{H2}$ g$_{cat}^{-1}$ h$^{-1}$) over 1000 min in a durability test (Supplementary Fig. 19). To evaluate the intrinsic rates, the differences in metal dispersion evaluated by STEM have to be taken into account (Supplementary Fig. 23). TOF values have been calculated based on the metal particle size distributions assuming fully exposed particle surfaces and equal activity of all surface atoms. These might not be fully realistic boundary conditions, but the systematic errors can be considered to be similar for all catalysts enabling a comparison with theoretical calculations regarding the trends among the four catalysts. Additional chemisorption combined with TPD were performed to determine the exposed metal surface based on the H$_2$ chemisorption capacity. Assuming the same adsorbate stoichiometry for all catalysts, the obtained metal dispersions follow the trend Fe/MgO (0.45%) ≈Fe$_{0.75}$Co$_{0.25}$/MgO (0.50%) ≤ Fe$_{0.5}$Co$_{0.5}$/MgO (0.65%) < Co/MgO (1.3%), which is similar to the trend obtained from the TEM particle size distribution: Fe/MgO (3.8%) ≥Fe$_{0.75}$Co$_{0.25}$/MgO (3.0%) ≤ Fe$_{0.5}$Co$_{0.5}$/MgO (3.2%) < Co/MgO (6.0%) (Supplementary Figs. 23 and 24). However, the chemisorption-derived dispersion are approximately six times lower than the ones obtained from the particle size distribution. The reasons could be oxide coverage not accounted for in the TEM evaluation, pre-mature removal of adsorbed hydrogen during the purging before TPD, and/or uncertainties in adsorbate stoichiometry. Therefore, while the general relative trend among the catalysts can be confirmed by chemisorption, we will base the following discussion

of the absolute values on the TEM-derived dispersions regarding them as a lower limit for the corresponding TOF values, which would be even higher if derived from the chemisorption data.

Importantly, the TOFs of these four catalysts at 500 °C show the same volcano-shaped variation with increasing Co content like the mass-normalized rate data as illustrated in Fig. 4b, but the superiority of the bimetallic catalysts is much clearer. Interestingly, both bimetallic catalysts $Fe_{0.75}Co_{0.25}$/MgO and $Fe_{0.5}Co_{0.5}$/MgO show almost the same intrinsic activity (TOF ~ 0.26 ± 0.01 s$^{-1}$) for ammonia decomposition. However, Co/MgO showed a much smaller TOF of ~0.13 s$^{-1}$, which is comparable to Fe/MgO (~0.15 s$^{-1}$). From this, we conclude that already small amounts of Co promote the monometallic Fe catalysts very efficiently by suppressing nitride formation. Increasing the Co content towards monometallic Co catalysts does not lead to a further improvement in the intrinsic activity. The apparent activation energy ($E_a$) was calculated from Arrhenius plots of these four catalysts (Supplementary Fig. 20) and was found to be slightly lower for Fe/MgO and Co/MgO compared to the two bimetallic FeCo catalysts in accordance with the volcano-shaped trend. As the difference was not large (< 13 kJ mol$^{-1}$) and hardly significant, it can be assumed that ammonia decomposition operates via a similar reaction mechanism for iron- and cobalt-monometallic or bimetallic catalysts in line with the literature reporting that the rate-determining step is the nitrogen desorption step on Fe- and Co-based catalysts[24]. However, the emerging volcano-shape trend in activation energy can be related to the difference in nitrogen binding energy[26], as we will show through the use of DFT calculations. Regarding the very similar catalysts $Fe_{0.5}Co_{0.5}$/MgO and Co/MgO, we also measured the activation energy of in a gas stream with a higher concentration of 10% NH$_3$, which resulted in a by around 20 kJ mol$^{-1}$ lower $E_a$ of Co/MgO in agreement with a low nitrogen binding energy for Co (Supplementary Fig. 21).

The reaction mechanism of ammonia synthesis has been investigated using DFT calculations[36]. Importantly, through the use of scaling relations for the adsorption energies of intermediates[37] and transition states[38], activity volcanoes could be established where the theoretical turnover frequency is plotted as a function of the nitrogen binding energy to the steps of transition metal surfaces[36,39]. Boisen et al. established an activity volcano for ammonia decomposition as a function of the reaction energy of dissociative N$_2$ adsorption and showed that this differs from that usually obtained for ammonia synthesis as the conditions in terms of temperature and pressures are different (Fig. 4c shows their original work at 500 °C, 1 bar, and 20% NH$_3$)[26]. In their work, they developed the concept of combining strong and weak binding elements to nitrogen, that we follow in this work. They suggested CoMo as an attractive bimetallic combination for ammonia decomposition and they and others[40] found experimentally that Co$_3$Mo$_3$N nitride was formed in the bulk and is the active phase, which is not the case in the FeCo catalysts of this work, which remain unnitrided.

We employed DFT to calculate the nitrogen binding energies on the surfaces of the four catalysts used herein and interpreted them in the light of the volcano established by Boisen et al. Using our methodology we obtained nitrogen binding energies that differ only slightly from those reported in the original work (for Co(0001) we calculated −0.16 eV compared to −0.18 eV)[26]. An activity volcano using the experimentally obtained TOFs (normalized to the metal surface area) as a function of the calculated nitrogen binding energies for various Fe and FeCo surfaces is shown in Fig. 4d, using the same x-axis as in 4c. As can be seen from Fig. 4d, our data suggests a volcano type behavior analogous to that predicted by Boisen et al. Note, that the data for Fe(210) (strong binding left side of the volcano) is only hypotheticalnce under ammonia decomposition conditions, the monometallic Fe/MgO catalysts were nitridated to form the stable bulk Fe$_3$N, as suggested by *operando* XAS, ex situ XRD, and XES (Fig. 2b–d and Supplementary Fig. 10). This moves the nitrogen binding energy to the weak binding side and explains the low activity observed for our iron

catalyst. If nitridation could be avoided, Fe(210) should exhibit a higher activity as evident from Fig. 4c, d. Co(0001) has also a rather weak nitrogen binding energy and is hence a poor catalyst for both, ammonia synthesis and decomposition. Alloying iron with cobalt, however, leads to surfaces with nitrogen binding energies that are close to the top of the theoretical ammonia decomposition volcano. This is also the outcome of our experimental efforts that show that FeCo alloys have a higher activity than both, Fe$_3$N and Co. The effect of alloying iron with cobalt is thus twofold, (1) suppression of nitridation and (2) weakening of the nitrogen binding energy.

In summary, we have found that the high-loading alloyed FeCo bimetallic catalysts with "intermediate microstructure" between supported and bulk catalysts synthesized by our spinel approach through co-precipitation could efficiently release H$_2$ from NH$_3$ decomposition. In-depth and complementary characterization uncovered the nitridation of Fe from surface to bulk under ammonia decomposition, which is different from ammonia synthesis. DFT calculations provided further evidence for the nitridation of Fe and revealed that Fe$_3$N surfaces exhibit too weak binding energies and are thus less active. By alloying Fe with Co, this nitridation could be suppressed and the nitrogen binding energy is additionally influenced such that the binding energies move closer to the top of the activity volcano, leading to a highly active and stable catalytic performance. This work also indicates that alloying Fe by other metals with weak nitrogen adsorption energy provides a simple and general approach to fabricating a highly active and unnitrided catalyst for ammonia decomposition reaction.

## Methods

### Materials
For the synthesis of the catalyst precursors the following commercially available chemicals were used without further purification: Cobalt (II) nitrate hexahydrate (≥98% p.a., ACS, Carl Roth GmbH & Co. KG), iron (II) sulfate heptahydrate (≥99.5% p.a., ACS, Carl Roth GmbH & Co. KG), iron (III) nitrate nonahydrate (≥98% p.a., ACS, Alfa Aesar GmbH), magnesium nitrate hexahydrate (≥98%, ACS, Alfa Aesar GmbH), sodium carbonate (p.a., AppliChem GmbH) and sodium hydroxide (≥99%, VWR International BVBA).

### Catalyst synthesis
Mg(Fe$_{1-x}$Co$_x$)$_2$O$_4$ spinel catalysts were prepared by co-precipitation of LDHs/hydroxide in an automated laboratory reactor system (Optimax synthesis workstation, Mettler Toledo). During co-precipitation, the metal solution contained three equal concentrations of Mg$^{II}$/Fe$^{III}$/M$^{II}$ (M$^{II}$ = Fe, Co) of 0.266 mol L$^{-1}$, where the ratio of Fe$^{II}$:Co$^{II}$ determines the ratio Fe:Co in Mg(Fe$_{1-x}$Co$_x$)$_2$O$_4$. The aqueous 0.6 mol L$^{-1}$ NaOH and 0.09 mol L$^{-1}$ Na$_2$CO$_3$ solution serve as a precipitation agent. The pH during co-precipitation is 10.0 for Mg(Fe$_{0.75}$Co$_{0.25}$)$_2$O$_4$ as well as Mg(Fe$_{0.5}$Co$_{0.5}$)$_2$O$_4$ and is 10.5 for MgFe$_2$O$_4$ with an aging time of 24 h at 50°C. For MgCo$_2$O$_4$, Co$^{II}$ solution of 0.533 mol L$^{-1}$ and Mg$^{II}$ solution of 0.266 mol L$^{-1}$, the ratio of Co$^{II}$:Mg$^{II}$ is 2:1. 1.0 mol L$^{-1}$ NaOH solution serve as a precipitation agen. The pH during co-precipitation is 11.0 with an aging time of 1 h at 50°C.

After washing with water until the conductivity of the supernatant was below 100 μS cm$^{-1}$, then after drying, the precursors were calcined at 600°C for 3 h and further isothermally reduced in H$_2$ prior to the reaction.

### Catalyst characterization
Iron, cobalt and magnesium contents in the spinel samples were determined by atomic absorption spectroscopy (AAS) (Thermo Electron Corporation, M-Series). The sodium contents in the samples were determined by Inductively Coupled Plasma Optical Emission spectroscopy (ICP-OES) (Avio 200 von PerkinElmer). Thermogravimetric measurements (TG) were performed in a NETZSCH STA 449F3 thermal analyzer. In a corundum crucible ca. 50 mg of the LDH / hydroxide

precursors were heated in synthetic air (21% $O_2$ in Ar) from 30 °C to 1000 °C with a linear heating rate of 5 °C $min^{-1}$. $N_2$ adsorption-desorption experiments of the LDH and spinel precursors were conducted with a NOVA3000e setup (Quantachrome Instruments) at −196 °C after degassing the samples at 100 °C for 2 h in vacuum. BET (Brunauer Emmet Teller) surface areas were calculated from $p/p_0$ data between 0.05 and 0.3. Pore volumes and pore size distribution were determined by applying the BJH (Barrett-Joyner-Halenda) method. Powder X-ray diffraction (XRD) patterns of the LDH and spinel phases were recorded on a Bruker D8 Advance diffractometer in Bragg-Brentano geometry using a position-sensitive LYNXEYE detector (Ni-filtered CuKα radiation). A 2θ range from 5° to 90°, a counting time of 2.96 s and a step width of 0.01° was applied. The samples were dispersed with ethanol on a glass disc inserted in a round PMMA holder, which was subjected to a gentle rotation during scanning. Powder XRD patterns of the reduced catalysts were recorded on a STOE transmission diffractometer STADI P in Debye-Scherrer geometry at room temperature using a curved image-plate position sensitive detector ($R = 150$ mm, crystal monochromator filtered MoKα radiation). A 2θ range of $2 \times 70°$ and a step width of 0.001° was applied. The samples were transferred from the reactor into a glove box under argon, then pestled and filled into a capillary sample holder (0.2 mm Ø). After sealing the capillary with vacuum grease, the sample holder was discharged from the glove box and the plugged opening of the capillary was sealed by melting to prevent sample contamination with air. Environmental scanning electron microscope (SEM) studies of spinel pre-catalysts were carried out with a Quanta ESEM 400 FEG microscope, equipped with a FEI detector and an Apreo S LoVac microscope (Thermo Fisher Scientific) which is equipped with two backscattering electron detectors. All samples were surface coated with Au:Pd (80:20, 6-7 nm) prior to the measurements. The non-resonant Fe Kβ X-ray Emission Spectroscopy (XES) data were collected at beamline ID26 of the European Synchrotron Radiation Facility (ESRF), which operates at 6 GeV with 200 mA ring current. A Si(111) double-crystal monochromator was used upstream for energy selection and calibrated to the first inflection point of an Fe foil set to 7111.2 eV. The incident beam energy was set to 7800.0 eV (above the Fe K-edge at 7112 eV) to excite the sample non-resonantly. The beam size at the sample was 1.0 mm (H) × 0.1 mm (V) and the photon flux was ~1012 ph/s (without attenuation). The non-resonant XES spectra were collected with a 1 meter radius Johann spectrometer equipped with five Ge(620) crystal analyzers and an avalanche photodiode (APD) detector aligned on intersecting Rowland circles. The spectrometer was internally calibrated using the $K\beta_{1,3}$ emission line of $Fe_2O_3$ at 7060.6 eV. All samples were diluted in boron nitride (BN) and measured in a liquid helium cryostat operated at 20 K. All XES spectra were collected between 7020.0 eV and 7130.0 eV with a uniform step size of 0.25 eV and were normalized to a unit area of 1 over the $K\beta_{1,3}$ mainlines region between 7020.0 eV to 7080.0 eV. $H_2$-Temperature programmed reduction ($H_2$-TPR) experiments were performed in a BELCAT-B (BEL Japan, Inc.) catalyst analyzer at a linear heating rate of $\beta = 6$ °C $min^{-1}$ between room temperature and 1000 °C. This temperature was held for 15 min before cooling down. All samples were dried at 100 °C for 60 min in Ar (80 mL$_n$ min-1) prior to the experiments. To remove the $H_2O$ from the gas stream an in-line molecular sieve was used before reaching a built-in thermal conductivity detector. A flow rate of 80 mL$_n$ $min^{-1}$ and 7% $H_2$ in Ar ($H_2 \geq 99.999$%, Ar $\geq 99.999$%, Air Liquide) was applied for the reduction process. For all experiments with this catalyst analyzer, ~20 mg of the sample with a sieve fraction of 250−355 μm was prepared. A quantitative analysis of the $H_2$ consumption was achieved by integrating the TCD signal obtained by the reduction of three different amounts of commercial CuO for calibration. It was used since CuO undergoes a complete reduction to $Cu^0$. $H_2$-Temperature programmed desorption ($H_2$-TPD) experiments for obtaining information of metal dispersion were

performed in a BELCAT-II (BEL Japan, Inc.) at a linear heating rate $\beta = 6$ °C $min^{-1}$ between -76°C and 200 °C. All samples (around 20 mg for each sample) were isothermally reduced at 600 °C for 5 h in 7% $H_2$/Ar (80 mL$_n$ $min^{-1}$, $H_2 \geq 99.999$%, Ar $\geq 99.999$%, Air Liquide) prior to the experiments, and purged with argon during cooling to 150 °C. The atmosphere was switched back from Ar to $H_2$ (60 mL$_n$ $min^{-1}$) at 150 °C, then the sample was cooled to −76 °C and kept in $H_2$ for adsorption at −76 °C for 1 h. Afterwards, the sample was purged by Ar at −76 °C for 2 h to remove $H_2$ in gas phase and physiosorbed $H_2$. Then, the samples were heated in Ar (60 mL$_n$ $min^{-1}$) to 200°C and the concentration of $H_2$ in the exhaust was monitored by a mass spectrometry (QMG 220, Pfeiffer Vacuum GmbH). The chemisorption data were used for calculating the values of metal dispersion assuming a stoichiometric ratio of $H_2$ to iron and / or cobalt of 2.

**In situ XRD measurements**
The data were collected on a STOE theta/theta X-ray diffractometer (Cu $K_{\alpha1+2}$ radiation, secondary graphite monochromator, scintillation counter) equipped with an Anton Paar XRK 900 in situ reactor chamber. The gas feed was mixed by means of Bronkhorst mass flow controllers, using 7% $H_2$ in helium at a total flow rate of 100 mLn $min^{-1}$. Due to the low time resolution (ca. 10 h per scan), all XRD measurements were performed at 25 °C to avoid continuous reduction of the sample during the data collection ("quasi in situ"). The samples were reduced in an in situ chamber with a ramp rate of 6 °C $min^{-1}$ until the respective target temperature was reached, followed by fast cooling (20°C $min^{-1}$) and XRD measurement at 25 °C. Subsequently, the sample was heated again at 20 °C $min^{-1}$ up to the previous target temperature, where the ramp rate was changed to 6°C $min^{-1}$ again until the next target temperature was reached, followed again by rapid cooling and XRD measurement.

**STEM-EDS-map measurements**
STEM-EDS micrographs of the spinel catalysts and freshly reduced catalysts were generated, using a Thermo Scientific Talos F200X. The transmission electron microscope was equipped with a high brightness field emission gun (X-FEG) and 4 SDD EDX detectors, giving together a detection area of 0.9 sr. The electron beam energy was 200 keV and the beam current 50 pA. The point resolution of the microscope was 1.6 Å. In order to reduce artifacts induced by electron beam on the sample, the multiple frame approach was applied. This means that the electron beam was scanned across the region of interest several times, with short acquisition times ranging from 20 to 50 μs per pixel, and the signal was integrated later. To compensate for sample drift during EDS acquisition, Velox drift correction was applied by cross-correlation after each frame. The mono-metal / bi-metal particle size distribution was determined by the evaluation of at least 400 particles.

**Dispersion of mono- / bi-metal nanoparticles: calculation from STEM results**
With the known diameter ($d_i$) of the individual metal nanoparticles ($n_i$), as measured by STEM, the volume-area mean diameter ($d_{VA}$) was calculated according to Eq. (1). From this relation, one can easily calculate the metal dispersion ($D_{metal}$), which is defined by the ratio of surface atoms to the total number of atoms in the hemispherical metal particle ($V_{metal}$ = volume metal atom, $a_{metal}$ = surface area metal atom) as shown in Eq. (2).

$$d_{VA} = \frac{\sum_i n_i d_i^3}{\sum_i n_i d_i^2} \tag{1}$$

$$D_{metal} = 6 \frac{V_{metal}/a_{metal}}{d_{VA}} \tag{2}$$

## Kinetic measurements

Prior to the catalytic steady-state measurements, 20 mg of the spinel catalysts (250–355 μm) were isothermally reduced for 5 h at 600 °C (7% $H_2$/Ar, 80 $mL_n$ $min^{-1}$, $\beta = 6$ °C $min^{-1}$, Air Liquide, $H_2 \geq$ 99.999%, Ar $\geq$ 99.999%). Afterward, the freshly reduced samples were cooled down to 500 °C and nitridated for 5 h in a mixture of 3% $NH_3$ in He (40 mLn $min^{-1}$, Air Liquide, $NH_3 \geq$ 99.999%, He $\geq$ 99.9996%). The nitridation step was initially added for Fe/MgO catalyst to obtain stable catalytic activity and finally employed to all $Fe_{1-x}Co_x$/MgO catalysts to maintain comparability. Then steady-state catalytic ammonia decomposition measurements were conducted between 400 and 600 °C. Firstly, the samples were cooled down to 400 °C at a gas flow of 80 $mL_n$ $min^{-1}$ (3% $NH_3$/He) which was directed through the catalyst bed. After measuring at 400 °C, the samples were heated up stepwise to 425, 450, 475, 500, 550, 600 °C and held at each temperature for 3 h. A mass spectrometer (QMG 220, Pfeiffer Vacuum GmbH) was calibrated for $NH_3$, $N_2$ and $H_2$ taking helium as the internal standard to perform a quantitative analysis during catalysis. A Micro GC Fusion from Inficon was calibrated for $NH_3$, $N_2$ and $H_2$ to perform analysis. The mole and corresponding volume changes of the gas mixture due to reaction and its effect on the conversion data was neglected as it is limited to the only 3% $NH_3$ in the gas feed and the 97% of diluent He fraction remain. The $NH_3$ conversions of all catalysts at 600 °C (89–95%) are approaching, but remain below the equilibrium $NH_3$ conversion (~99%). The metal-mass-normalized $H_2$ production rates and the apparent activation energy ($E_a$) were calculated at low $NH_3$ conversion approaching differential reaction conditions. Calculations of the Weisz-Prater criterion and Mears' criterion confirm the absence of mass transfer limitations for this catalytic system as described in the Supplementary Information.

## *Operando* XAS measurements

Measurements were performed at the CAT part of the CAT-ACT beamline at the KIT light source in Karlsruhe, Germany[41]. The samples were typically diluted with boron nitride, pressed into pellets, crushed and mesh-sieved between 100 and 200 μm. For the measurements, the samples were loaded into a quartz capillary (1.5 mm o.d., 0.02 mm wall thickness, ~6 mm bed length) and mounted on top of a hot air blower[42]. The gas flow rate through the capillary was 50 $mL_n$ $min^{-1}$, and the outlet gas composition was monitored by a Pfeiffer Vacuum OmniStar GSD 320 mass spectrometer (example for on-line analysis in Supplementary Fig. 15). The X-ray absorption near edge structure (XANES) and the extended X-ray absorption fine structure (EXAFS) spectra were collected at the Fe K-edge (7112 eV) (for both $MgFe_2O_4$ and $Mg(Fe_{0.5}Co_{0.5})_2O_4$ pre-catalysts) and Co K-edge (7709 eV) (only for $Mg(Fe_{0.5}Co_{0.5})_2O_4$ pre-catalyst), in transmission mode with ion chambers as detectors. The following experiments were done on this sample after the initial loading: Reduction by 5% $H_2$ in He from 30-600 °C (with 6 °C/min ramp rate) and dwelling of 8 h, cooling down to 425 °C and exposure to 0.5% $NH_3$, ramping to 550 °C under $NH_3$ and dwelling of 2 h, cool down to 30 °C under $NH_3$ environment. XANES spectra and an extended energy range for the EXAFS spectra were collected and at the end of that stage. The collected data were normalized using the Athena code within the Demeter package (version 0.9.26)[43]. The EXAFS part of the spectra was obtained by converting the normalized data from energy space to k space and weighted by k, $k^2$, and $k^3$, then Fourier transformed in the selected k range (depending on data quality) with a Hanning window. The fitting of the data was then performed in R-space with Artemis software of the same package with selected structural models for all k-weighted datasets.

## DFT calculations

DFT calculations in this work were carried out using the Vienna Ab Initio Simulation Package (VASP)[44,45] in connection with the Atomic Simulation Environment (ASE)[46]. A plane-wave basis set with a cutoff energy of 450 eV, the projector augmented wave method (PAW)[47,48] and the Bayesian Error Estimation Functional with van der Waals correlations (BEEF-vdW)[49] exchange correlation functional were used. The lattice constants were optimized using an energy cutoff of 450 eV and a 9 x 9 x 9 (for Fe, FeN, $Fe_3N$ and FeCo), 5 x 5 x 5 (for $Fe_3Co$) Monkhorst Pack k-point sampling. The optimize lattice constants are given in Supplementary Table 6.

The kinetic energy cutoff for all slab calculations was 450 eV. The Fe(210), $Fe_3Co(210)$ and FeCo(210) systems were modeled by infinite slabs consisting of four layers thick $3 \times 2$ super-cells. Cobalt surfaces were modeled using four layer thick $4 \times 2$ large Co(0001) unit cell, with one row of Co atoms in the y-direction removed from the top layer creating A and B edges. Iron nitride was modeled using a four layer thick $Fe_3N(111)$ surface with 6Fe and 2 N atoms in each layer. All slab models were separated by > 15 Å in the z direction. In all calculations the bottom two layers were kept fixed at the bulk positions while the top two layers were allowed to relax during geometry optimizations. The convergence criterion for geometry optimizations was a maximum force of 0.01 eV/Å. The Brillouin zones were sampled using a $4 \times 4 \times 1$ (for Fe(210), $Fe_xCo(210)$, $Fe_3N(111)$) and $3 \times 6 \times 1$ (for Co(0001)) Monkhorst–Pack k-point grid for[50]. Spin polarization was considered in calculations. Adsorption energies of nitrogen were calculated relative to gas-phase $N_2$. Zero-point-energy corrections were calculated from vibrational analyses carried out in the harmonic approximation using a finite difference with a magnitude of 0.01 Å for the displacements. The coordinates of all structures are given in the Supplementary Information.

## Data availability

The data generated in this study are provided in the Supplementary Information/Source Data file. Source data are provided with this paper.

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

## Acknowledgements

S.C., M.B., L.K. and S.D. would like to thank the Federal Ministry of Education and Research, Germany (Bundesministerium für Bildung und Forschung, BMBF, Hydrogen flagship project: TransHyDE Forschungsverbund AmmoRef, FKZ 03HY203E and FKZ 03HY203A) for funding. L.K., A.W. and S.D. thank the Max Planck Society for funding. L.K. acknowledges Alexander von Humboldt Foundation for a post-doctoral fellowship and funding support. We thank the Institute for Beam Physics and Technology (IBPT) for the operation of the storage ring, the Karlsruhe Research Accelerator (KARA) and thank the KIT light source for provision of instruments at the CAT-ACT beamline of the

Institute of Catalysis Research and Technology (IKFT). We acknowledge Dr. Anna Zimina and Dr. Tim Prüßmann for their help during *Operando* XAS measurements. The XES experiments were performed on beamline ID26 at the European Synchrotron Radiation Facility (ESRF), Grenoble, France. We are grateful to Dr. Pieter Glatzel at the ESRF for providing assistance in using beamline ID26.

## Author contributions

S.C. and D.R. conducted material synthesis, basic charaterizations, catalytic performance tests and data analysis. J.J. and F.S. performed the DFT calculations and interpretation. F.-P.S. performed STEM-EDS-map measurements. S.C. conducted the metal dispersion evaluation and H2 chemisorption experiment. F.G. performed in situ XRD measurements and data analysis. S.N., D.R. and K.F.O. performed the XAS experiments with the assistance of D.E.D. L.K. and S.D. conducted the XAS data analysis. S.C., S.N., D.E.D., J.-D.G. and M.B. contributed to the discussion and data interpretation of XAS data. L.K., A.W. and S.D. performed the XES experiments and data analysis. A.R. synthesized and characterized the Co/MgO catalyst. J.W. performed additional XRD measurements for the revision. T.L. and R.S. contributed to the discussion and data interpretation. M.B. supervised the project. The first draft was written and revised by S.C. and M.B. All authors contributed to the discussion, manuscript writing and revision of the manuscript.

## Funding

## Competing interests

The authors declare no competing interests.
