## [Peer Review File · Nature Communications]

Highly-loaded Bimetallic Iron-Cobalt Catalysts for Hydrogen Release from AmmoniaREVIEWER COMMENTS

Reviewer #1 (Remarks to the Author):

Shilong Chen et al. studied hydrogen evolution on the iron-based catalyst through ammonia decomposition. The interesting features of this work include the novel approach to catalyst fabrication, namely, from the MgFe_2O_4 or $\text{MgCo}_x\text{Fe}_{(1-x)}\text{O}_4$ spinel catalyst precursor. The resulting materials showed attractive low-temperature performance (at 500°C for example) but the overall activity at higher temperature is not cutting-edge among the data reported previously on a variety of Fe-based catalysts. Nevertheless, the characterizations are highly complementary and in-depth by employing the in-situ techniques, providing the new insights into the atomic level description of local structure of catalyst surface and bulk, and the resulting electronic effect and catalytic behavior for the target reaction. The authors put forward a useful tactic for constructing an efficient Fe-based catalyst by alloying another non-precious metal with appropriate nitrogen binding energy. My overall impression is that the work is informative and publishable in this journal; on the other hand, some issues need to be addressed or reconsidered.

1. Figure S17: the ammonia conversion increases rapidly with increasing temperature (from 400 to 500°C), however, the increase in conversion slows down with further increasing temperature from 500 to 600°C. Since the fed ammonia is highly diluted (3% in He), why such a fraction of ammonia cannot be completely converted at 600°C? Is there an equilibrium control or a mass-transfer limitation existing for the reaction? More discussion should be provided on this point.
2. Considering the estimation of the TOF values, how to accurately determine the exposure of the metal surface? The authors mentioned the metal dispersion in terms of the average particle size derived from the STEM measurements. However, in the present case, there is a special situation, the metal surfaces are partially covered by a number of small oxide entities (TEM characterizations), and the size and distribution of oxide species are rather random, thus the assumption that the metal surface is fully exposed looks unreasonable. Bear also in mind that the standard error in the listed mean metal particle size appears quite large, that is to say, using the average particle size to estimate the metal dispersion in this case is questionable. Is there any experimental approach such as selective chemisorption of suitable probe molecule (NO_x), to determine the exposed (uncovered) metal surface for TOF calculation? In addition, what is the ammonia conversion level employed for the TOF calculation?
3. Regarding the structural modeling used for DFT simulation, are these illustrated surface structures unreconstructed or reconstructed? In the case of Fe-Co alloying, how about the segregation effect on the surface composition of the bimetallic catalysts? Should such an effect be significant for structural modeling and DFT calculation?
4. The authors mentioned the un-nitrided catalyst is favorable for ammonia decomposition. However, I noticed that the freshly reduced were actually subjected to a nitridation process (500°C for 5 h) before the steady-state reaction, why such a process was specifically adopted?
5. I did not see the long-term durability test conducted on a representative catalyst in the current study.

Reviewer #2 (Remarks to the Author):

Fast rate hydrogen production from ammonia decomposition catalyzed by non-precious metal is a practical need. It is also a scientific challenge with growing research interests. Fused Fe has been used in H-B process, but Fe is relatively less active for NH_3 decomposition owing to the facile formation of

nitride under NH₃ decomposition condition (temperatures < 650C). This work attempted to tune the Fe-N bond strength via forming Fe-Co alloy, which is successful in that higher activity was achieved and no nitride was formed.

This work is a nice combination of experimental and theoretical efforts. The catalysts have been well prepared and characterized by operando techniques. I will suggest accept this work after minor revision.

Alkalis have been discussed as electronic promoter even if its content is not significant. In preparing the catalysts the authors used NaOH or Na₂CO₃ to precipitate precursors. It would be good to provide elemental analysis on Na content.

The activity difference is not significant between Fe-Co/MgO and Co/MgO under the condition applied, so do the apparent activation energies. Perhaps varying flow rate or NH₃ concentration would distinguish those catalysts better.

The Co-containing catalysts are more active so that the NH₃/H₂ ratio would favor metallic Fe instead of Fe₃N. In other words, if physically mixing Fe/MgO and Co/MgO for ammonia decomposition, Fe in Fe/MgO perhaps is also in metallic state because the atmosphere (NH₃/H₂) is in favor of. It would be arguable that Co-Fe as catalyst or Co works alone. More discussion on the alloy catalyst would be helpful.

Reviewer #3 (Remarks to the Author):

This work an MgFeO solid solution, promoted with cobalt as a catalyst for ammonia decomposition. The authors claim that the high activity is associated to the lack of nitridation suppressed by the presence of cobalt, following a well-reported approach consisting of combining metals with weak and strong nitrogen binding energy. In my opinion, this paper contains a number of inconsistencies and thus, I don't think it should be published in Nature Comms.

1. Promoted iron-based catalysts are active for ammonia decomposition at high temperatures, as the one presented here as reported in a number of recent reviews. The authors do not report low temperature activity and thus, I find the title misleading.
2. There is a considerable body of literature following the approach of combining metals with weak and strong nitrogen binding energy to enhance the activity of non-noble metal catalysts, mainly iron and cobalt-based, so this work presents limited novelty. See for example: 10.1016/j.ijhydene.2018.07.085, 10.1016/j.ijhydene.2014.06.081, 10.1016/j.apcatb.2020.119405 in addition to the references in the manuscript 14-18 about iron-based alloys
3. The authors report TOF numbers based "on the metal particle size distribution assuming fully exposed particle surfaces and equal activity of all surface atoms". If I understand this correctly, they consider that only surface atoms are involved in the reaction and thus, they report considerably TOF numbers that if considering the whole metal amount (which is standard in the literature). If I am correct, I consider this misleading and wrong.
4. The effects on particle sizes presented in Figure S20 should be discussed in the main text as the authors should not discard the effect of size on activity
5. Following previous comment, it is not clear what the catalyst in Fig 1 e STEM figures is – is it after reduction at 600C at 5 h?
6. The authors should compare their catalytic activity to other iron-based catalysts in the literature as well as Ru-based catalyst, currently being the state-of-the art system for ammonia cracking.

7. Figure 1d and Figure 3b show the XRD-based identification of species in the (MgFe)O and (MgFeCo)O catalysts showing different species in both catalysts – which one are the active ones?
8. I also find confusing that authors talk about Fe/MgO catalyst when they actually have $\text{Fe}_{(1-x)}\text{Mg}_x\text{O}$ solid solutions. Again this is misleading.
9. The operando SAX experiments in Figure 2 are done under fully reduced conditions which are different to Fig 1d-XRD. If nitride formation happens at 55C – why is low temperature activity low? Do the authors have results for a second run (e.g. after decreasing temperature) to demonstrate that indeed nitride formation is responsible of low activity?
10. Regarding the experimental method – how do the authors consider the change in the number of moles during the reaction for the calculation of conversion values?

Response to reviewer comments on manuscript NCOMMS-23-26180-T

“Making Iron Active: Highly-loaded Bimetallic Iron-Cobalt Catalysts for Hydrogen Release from Ammonia”

Reviewer #1:

Comments:

Shilong Chen et al. studied hydrogen evolution on the iron-based catalyst through ammonia decomposition. The interesting features of this work include the novel approach to catalyst fabrication, namely, from the $MgFe_2O_4$ or $MgCo_xFe_{(1-x)}O_4$ spinel catalyst precursor. The resulting materials showed attractive low-temperature performance (at 500oC for example) but the overall activity at higher temperature is not cutting-edge among the data reported previously on a variety of Fe-based catalysts. Nevertheless, the characterizations are highly complementary and in-depth by employing the in-situ techniques, providing the new insights into the atomic level description of local structure of catalyst surface and bulk, and the resulting electronic effect and catalytic behavior for the target reaction. The authors put forward a useful tactic for constructing an efficient Fe-based catalyst by alloying another non-precious metal with appropriate nitrogen binding energy. My overall impression is that the work is informative and publishable in this journal; on the other hand, some issues need to be addressed or reconsidered.

Authors reply: We very much appreciate the positive comment of the reviewer and thank for her / his suggestions for improving the quality of the manuscript. Our point-by-point response is given below.

Comment 1): *Figure S17: the ammonia conversion increases rapidly with increasing temperature (from 400 to 500oC), however, the increase in conversion slows down with further increasing temperature from 500 to 600oC. Since the fed ammonia is highly diluted (3% in He), why such a fraction of ammonia cannot be completely converted at 600oC? Is there an equilibrium control or a mass-transfer limitation existing for the reaction? More discussion should be provided on this point.*

Authors reply: The equilibrium NH_3 conversion from 500 to 600°C is larger than 99% and thus very close to full conversion (Figure S17, Supplementary Information (SI)). For our four catalysts, the NH_3 conversion at 600°C is in the range from 89 to 95%, which is lower than the equilibrium NH_3 conversion, but slowly approaching it. As pointed out by the reviewer, measurements at such conditions may be affected by contributions beyond the intrinsic forward reaction rate and show a sigmoidal approach to equilibrium. We would like to emphasize that the kinetic evaluation discussed in our paper, is based on a lower reaction temperature of 500 °C, where the measurements are far away from the equilibrium and thus can be used to conclude on the intrinsic catalytic properties.

To estimate if possibly mass-transfer limitations are responsible for the sigmoidal behavior at high temperatures, the Weisz-Prater criterion / Mears' criterion for mass transfer limitations for this catalytic system has been calculated, where C_{wp} was found to be smaller than 1 as well as C_M smaller than 0.15. These calculations thus confirmed that the catalytic microreactor system for ammonia decomposition does not suffer from mass-transfer limitations. Therefore, we conclude that the NH_3 conversion values of these four catalysts at 600°C (89-95%) are not affected by mass transfer limitation, but by other aspects due to close proximity to equilibrium (~99%).

This information has been added in the revised Supplementary Information (page S5 and S20, SI) as the following part:

2. Weisz-Prater criterion / Mears' criterion calculations for mass transfer limitations for ammonia decomposition on Fe_{1-x}Co_x/MgO catalysts

2.1. Mass transfer limitations: Internal diffusion, Weisz-Prater Criterion²

The absence of internal mass transfer limitations can be verified by a Weisz-Prater criterion C_{WP} lower than 1.

$$C_{WP} = \frac{r_{abs}\rho_c R^2}{D_{eff}C_{As}} < 1$$

r_{abs} = observed maximum reaction rate, mol kg_{cat}⁻¹ s⁻¹ (here, the maximum NH₃ reaction rate at 95.0% of NH₃ conversion over Fe_{1-x}Co_x/MgO catalyst was used)

R = catalyst particle radius, m

ρ_c = solid catalyst density, kg m⁻³

D_{eff} = effective gas-phase diffusivity, m² s⁻¹

C_{As} = gas concentration of NH₃ at the external surface of the catalyst, mol m⁻³

Hence, $C_{wp} = \{[0.0777 \text{ mol kg}_{cat}^{-1} \text{ s}^{-1}] \times [4400 \text{ kg m}^{-3}] \times [2 \times 10^{-4} \text{ m}]^2 / \{[2.76 \times 10^{-5} \text{ m}^2 \text{ s}^{-1}] \times [1.23 \text{ mol m}^{-3}]\}$
= 0.40 < 1

Therefore, the effect of internal mass transfer limitation on the catalytic measurements of ammonia decomposition can be neglected.

2.2. Mass transfer limitations: External diffusion, Mears' criterion^{2,3}

The absence of external mass transfer limitations can be verified by a Mears' criterion C_M lower than 0.15:

$$C_M = \frac{r_{abs}\rho_b R n}{k_c C_{Ab}} < 0.15$$

r_{abs} = observed maximum reaction rate, mol kg_{cat}⁻¹ s⁻¹ (here, the maximum NH₃ reaction rate at 95.0% of NH₃ conversion over Fe_{1-x}Co_x/MgO catalyst was used)

ρ_b = bulk density of catalyst bed, kg m⁻³

R = catalyst particle radius, m

n = reaction order, using 0.6 as the maximum value for NH₃ reaction order in ammonia decomposition

k_c = external mass transfer coefficient, m s⁻¹

C_{ab} = bulk gas concentration of NH₃, mol m⁻³

Hence, $C_m = \{[0.0777 \text{ mol kg}_{cat}^{-1} \text{ s}^{-1}] \times [500 \text{ kg m}^{-3}] \times [2 \times 10^{-4} \text{ m}]^2 \times 0.6 / \{[0.316 \text{ m s}^{-1}] \times [1.23 \text{ mol m}^{-3}]\}$
= 1.20 × 10⁻² < 0.15

Therefore, the effect of external mass transfer limitation on the catalytic measurements of ammonia decomposition can be neglected

Figure S17 Steady-state NH₃ conversion of the four Fe_{1-x}Co_x/MgO catalysts during ammonia decomposition. The yellow curve represents the equilibrium NH₃ conversion of NH₃ at different temperatures under 1 bar, NH₃ (g) ↔ 1/2 N₂ (g) + 3/2 H₂ (g), ΔrH⁰_m (NH₃ (g), 298 K) = 46.2 kJ mol⁻¹, ΔrG⁰_m (NH₃ (g), 298 K) = -16.63 kJ mol⁻¹, ΔCp = 25.46 - 0.01833T + 205000T⁻².

The NH₃ conversion values of these four catalysts at 600°C (89-95%) are under the equilibrium NH₃ conversion (~99%).”

Comment 2): Considering the estimation of the TOF values, how to accurately determine the exposure of the metal surface? The authors mentioned the metal dispersion in terms of the average particle size derived from the STEM measurements. However, in the present case, there is a special situation, the metal surfaces are partially covered by a number of small oxide entities (TEM characterizations), and the size and distribution of oxide species are rather random, thus the assumption that the metal surface is fully exposed looks unreasonable. Bear also in mind that the standard error in the listed mean metal particle size appears quite large, that is to say, using the average particle size to estimate the metal dispersion in this case is questionable. Is there any experimental approach such as selective chemisorption of suitable probe molecule (NO_x), to determine the exposed (uncovered) metal surface for TOF calculation? In addition, what is the ammonia conversion level employed for the TOF calculation?

Authors reply: We thank for the reviewer pointing this out. Based on the STEM characterizations of the Fe_{1-x}Co_x/MgO catalysts, the large metal particles indeed are in contact to smaller oxide particles forming an interface, which will cover a fraction of total metal surface and limits the exposed surface areas of metal particles. Note that we determined the metal / alloy dispersion by using volume-area mean diameter / mean surface diameter ($d_{VA} = \sum_i n_i d_i^3 / \sum_i n_i d_i^2$), which considers in the whole range of the metal / alloy particles. This limits the effect of the indeed large standard error of the mean particle size ($d_{mean} = \sum_i n_i d_i / \sum_i n_i$) on the results.

The above-mentioned fraction of oxide coverage is difficult to determine accurately, but based on the STEM investigation, we have no reason to believe that this fraction will differ substantially between the four catalysts evaluated, because they all were synthesized from the same precursor structure with the same molar metal-to-oxide ratio and the same general microstructure. We thus agree with the reviewer that the oxide coverage will have an effect on the metal dispersion, but we conclude that it will be similar for all catalysts and will not affect the trend among the investigated samples. Note that the consideration of the dispersion was not necessary to see the volcano-like trend, which was already detected in the weight-based catalyst performance data, but it serves rather to further clarify this trend.

Nevertheless, we agree that the determination of the metal dispersion is rather an estimate of the highest exposed metal surface area, and thus gives rise to a lower limit of the TOF values. In order to provide an alternative method of estimation of the metal dispersion, we have conducted H₂ chemisorption on the four catalysts. This method does not rely on the assumption of similar microstructure, but on similar chemisorption stoichiometry on the bi-metallic surfaces of varying composition and thus should also be seen rather as an estimation for the dispersion.

Thus, H₂ chemisorption was combined with TPD assuming the H₂ chemisorption capacity to be a measure for the exposed metal surface area. The H₂ chemisorption experiments for Fe_{1-x}Co_x/MgO catalysts were done at -76°C to ensure the saturated adsorption. In the H₂-TPD, the trend for the amount of adsorbed H₂ is quite clear, following the order of Co/MgO two times higher than the rest catalysts (Figure S24a, SI), which leads to the dispersion of metal: Co/MgO (1.3%), Fe_{0.5}Co_{0.5}/MgO (0.65%), Fe_{0.75}Co_{0.25}/MgO (0.5%), and Fe/MgO (0.45%). This result confirms the trend previously determined based on the evaluated particle size distribution. However, the values of dispersion from H₂-TPD experiments are ~6 times lower than the one obtained from particle size distribution and this difference seems to too large to be solely due to the coverage with oxide particles. This discrepancy could also be due to uncertainties in the chemisorption stoichiometry and/or due to pre-mature desorption during the purging step before the H₂-TPD experiment.

In summary, we have used two independent methods to estimate the metal dispersion and found the same trend. We keep the dispersion evaluated from the particle size distribution for quantification of the TOF values considering this a lower limit of the TOFs.

The information and discussion have been added in the revised manuscript and Supplementary Information as follows (page 14f, and page S25):

“Additional chemisorption combined with TPD were performed to determine the exposed metal surface based on the H₂ chemisorption capacity. Assuming the same adsorbate stoichiometry for all catalysts, the obtained metal dispersions follow the trend Fe/MgO (0.45%) ≈ Fe_{0.75}Co_{0.25}/MgO (0.50%) ≤ Fe_{0.5}Co_{0.5}/MgO (0.65%) < Co/MgO (1.3%), which is similar to the trend obtained from the TEM particle size distribution: Fe/MgO (3.8%) ≥ Fe_{0.75}Co_{0.25}/MgO (3.0%) ≤ Fe_{0.5}Co_{0.5}/MgO (3.2%) < Co/MgO (6.0%) (Figure S23 and S24). However, the chemisorption-derived dispersion are approximately six times lower than the ones obtained from the particle size distribution. The reasons could be oxide coverage not accounted for in the TEM evaluation, pre-mature removal of adsorbed hydrogen during the purging before TPD, and/or uncertainties in adsorbate stoichiometry. Therefore, while the general relative trend among the catalysts can be confirmed by chemisorption, we will base the following discussion of the absolute values on the TEM-derived dispersions regarding them as a lower limit for the corresponding TOF values, which would be even higher if derived from the chemisorption data.”.

and “

Figure S24 H₂ desorption rate of these four catalysts during TPD after H₂ chemisorption at -70°C, the purple curve indicated the temperature during TPD (a). The metal dispersion obtained from adsorbed H₂ molecules during TPD (b).

After H₂ chemisorption on the catalysts at -76°C for 1h, the TPD in Ar was conducted at a heating rate of 6 K min⁻¹. All the Co-containing catalysts showed two clear peaks of H₂ desorption with the first feature at 10-18°C and the second feature at 83-90°C. The Fe/MgO only showed one broad peak of H₂ desorption at 80°C.”

Comment 3): Regarding the structural modeling used for DFT simulation, are these illustrated surface structures unreconstructed or reconstructed? In the case of Fe-Co alloying, how about the segregation effect on the surface composition of the bimetallic catalysts? Should such an effect be significant for structural modeling and DFT calculation?°

Authors reply: We would like to thank reviewer for the comment so that we could additionally clarify structures used.

The surface structures used are unreconstructed, meaning that the Fe₃Co(210) edge was built from the Fe₃Co bulk, similarly the FeCo(210) edge from the FeCo bulk and the Fe₃N(111) surface from ε-Fe₃N bulk. However, the 210 edge of the stoichiometric FeCo and Fe₃Co have two types of terminations: for FeCo(210) – a Co rich (66.7% Co) and a Fe rich (66.7% Fe phase). The N binding energies at these two terminations are: -0.34 eV for the Co rich termination and -0.74 eV for the Fe rich termination.

In order to test segregation effect on alloy stability in vacuum and with N adsorbed, we constructed 210 slabs with both, Co terminated edges (each with 66.7% Co) – with Fe segregation into the sublayer – and Fe terminated edges – with Co segregation into the sublayer. The process of Co segregation to the surface in the vacuum is +0.22 eV endothermic. The N binding on such a Co segregated surface is -0.13 eV referenced to the initial, unreconstructed structure and -0.35 eV referenced to the segregated structure.

The process of Fe segregation to the surface in the vacuum is +0.45 eV endothermic. The N binding on such a Fe segregated surface is, however, -0.31 eV referenced to the initial, unsegregated structure and -0.76 eV, referenced to the Fe surface segregated system.

From these calculations we can conclude that the most stable structure is the stoichiometric, unreconstructed FeCo(210) with N adsorbed at the Fe rich termination. For this reason, the Fe rich terminations for FeCo(210) and Fe₃Co(210) were chosen as adsorption surfaces.

We have accordingly changed the Supplementary Information by adding the following part (page S27):

“The alloy and nitride structures used in this work are unreconstructed surfaces obtained by truncation of the bulk structure. However, the 210 edge of the stoichiometric FeCo has two types of terminations: a Co rich (66.7% Co) and a Fe rich (66.7% Fe) surface. The N binding energies on these two terminations differ by about 0.4 eV in favor of the Fe rich terminations. Additional Co and Fe segregation to the surface were tested with and without N adsorption. We found that both, the segregation of Co and Fe are unfavorable by +0.2 eV and +0.45 eV, respectively. We therefore chose Fe rich terminations for the stoichiometric FeCo(210) and Fe₃Co(210) as adsorption sites.”

Comment 4): The authors mentioned the unnitrided catalyst is favorable for ammonia decomposition. However, I noticed that the freshly reduced were actually subjected to a nitridation process (500oC for 5 h) before the stead-state reaction, why such a process was specifically adopted?

Authors reply: We thank the reviewer for pointing this out. For pure Fe/MgO catalyst, this nitridation step were added to obtain a stable catalytic activity during reaction, as otherwise the NH₃ consumption due to nitridation would overlap with the catalytic NH₃ conversion in the initial period of the measurement. Then to test whether nitride formation would occur on FeCo catalysts under the same conditions, we added this pre-treatment step for Co catalysts to maintain comparability. We have added this information to the revised manuscript (page 19) as the following: “The nitridation step was initially added for Fe/MgO catalyst to obtain stable catalytic activity and finally employed to all Fe_{1-x}Co_x/MgO catalysts to maintain comparability.”

Comment 5): I did not see the long-term durability test conducted on a representative catalyst in the current study.

Authors reply: According to the reviewer’s suggestion, we did the stability test of the most active catalyst Fe_{0.5}Co_{0.5}/MgO. The activity of Fe_{0.5}Co_{0.5}/MgO for ammonia decomposition at 500°C kept stable for more than 1000 min.

We added the new figure (Figure S19, page S21) in the Supplementary Information and the additional texts in the revised manuscript (page 14) as the following: “Furthermore, the Fe_{0.5}Co_{0.5}/MgO catalyst kept a stable reaction rate at 500 °C (~0.30 mol_{H2} g_{cat}⁻¹ h⁻¹) over 1000 min in a durability test (Figure S19, SI).” and “

Figure S19 Stability test of Fe_{0.5}Co_{0.5}/MgO catalyst for ammonia decomposition at 500°C (3%NH₃, 80 ml min⁻¹).”

Reviewer #2:

Comments:

Fast rate hydrogen production from ammonia decomposition catalyzed by non-precious metal is a practical need. It is also a scientific challenge with growing research interests. Fused Fe has been used in H-B process, but Fe is relatively less active for NH₃ decomposition owing to the facile formation of nitride under NH₃ decomposition condition (temperatures < 650C). This work attempted to tune the Fe-N bond strength via forming Fe-Co alloy, which is successful in that higher activity was achieved and no nitride was formed. This work is a nice combination of experimental and theoretical efforts. The catalysts have been well prepared and characterized by operando techniques. I will suggest accept this work after minor revision.

Authors reply: We thanks a lot the reviewer for her / his very positive comment. Our point-by-point response is given below.

Comment 1): *Alkalis have been discussed as electronic promoter even if its content is not significant. In preparing the catalysts the authors used NaOH or Na₂CO₃ to precipitate precursors. It would be good to provide elemental analysis on Na content.*

Authors reply: Thanks for this comment. Indeed, NaOH and / or Na₂CO₃ were used as precipitating agents in the synthesis of the precursors, and the precursors were always washed with the water until the conductivity of the supernatant was below 100 μS cm⁻¹. As suggested by the reviewer, we have now also checked the efficiency of this treatment and measured the Na content of all the four samples by ICP-OES. The Na concentration of the four samples show low value around ~0.20 wt.% which is similar for all samples. Therefore, exclude differences in the sodium content as decisive factors for the difference of the catalytic activity for these four samples. This information has been added in the revised Supplementary Information (page 3 and 8, SI), including in experimental methods: “**The sodium content in the samples were determined by Inductively Coupled Plasma Optical Emission spectroscopy (ICP-OES) (Avio 200 von PerkinElmer)**” and in synthesis part: “**The residual sodium content of all samples as measured by ICP-OES was around 0.20 ± 0.01 wt.% for the calcined samples**”.

Comment 2): *The activity difference is not significant between Fe-Co/MgO and Co/MgO under the condition applied, so do the apparent activation energies. Perhaps varying flow rate or NH₃ concentration would distinguish those catalysts better.*

Authors reply: Indeed, the NH₃ conversion of Fe_{0.5}Co_{0.5}/MgO and Co/MgO are similar and the apparent activation energy shows only little difference. Following the comment from the reviewer, we varied the NH₃ concentration from 3% to 10% to additionally evaluate the catalytic activity of these two catalysts. However, the difference in NH₃ conversion between these two catalysts are still not very big (Figure S21, SI) and similar to the previous activity results (Figure S17, SI). Compared to the apparent activation energy measured in the gas mixture of 3% NH₃, Co/MgO showed lower value E_a of 86.1 kJ mol⁻¹ while Fe_{0.5}Co_{0.5}/MgO showed similar E_a of 106.8 kJ mol⁻¹ in the gas mixture of 10% NH₃. The difference in E_a for the Co/MgO catalyst might be due to some change of the reaction mechanism and surface coverages

under different concentrations of NH_3 . However, in the gas mixture of 10% NH_3 , the difference in E_a between $\text{Fe}_{0.5}\text{Co}_{0.5}/\text{MgO}$ and Co/MgO is around 20 kJ mol^{-1} , which agrees well with the lower nitrogen binding energy of Co.

The figure and information has been added in the revised manuscript (page 15) and Supplementary Information (page S22) as follows: “Regarding the very similar catalysts $\text{Fe}_{0.5}\text{Co}_{0.5}/\text{MgO}$ and Co/MgO , we also measured the activation energy of in a gas stream with a higher concentration of 10% NH_3 , which resulted in a by around 20 kJ mol^{-1} lower E_a of Co/MgO in agreement with a low nitrogen binding energy for Co (Figure S21, SI).”, and “

Figure S21 NH_3 conversion of the $\text{Fe}_{0.5}\text{Co}_{0.5}/\text{MgO}$ and Co/MgO catalysts in ammonia decomposition in a gas mixture with 10% NH_3/Ar at a flow rate of 80 ml min^{-1} (a). Arrhenius plots of $\text{Fe}_{0.5}\text{Co}_{0.5}/\text{MgO}$, and Co/MgO catalysts for ammonia decomposition in the gas mixture of 10% NH_3/Ar with flow rate of 80 ml min^{-1} under differential reaction conditions (b).

The difference in NH_3 conversion for $\text{Fe}_{0.5}\text{Co}_{0.5}/\text{MgO}$ and Co/MgO was very similar in 3% NH_3/Ar . (see main text). If measured in 10% NH_3 , the two catalysts are still very similar. Compared to 3% NH_3 , however, the apparent activation energies Co/MgO showed lower E_a of 86.1 kJ mol^{-1} while $\text{Fe}_{0.5}\text{Co}_{0.5}/\text{MgO}$ showed an E_a of $106.8 \text{ kJ mol}^{-1}$ in the gas mixture of 10% NH_3 , which is similar to the value obtained at 3% NH_3 .”

Comment 3): The Co-containing catalysts are more active so that the NH_3/H_2 ratio would favor metallic Fe instead of Fe_3N . In other words, if physically mixing Fe/MgO and Co/MgO for ammonia decomposition, Fe in Fe/MgO perhaps is also in metallic state because the atmosphere (NH_3/H_2) is in favor of. It would be arguable that Co-Fe as catalyst or Co works alone. More discussion on the alloy catalyst would be helpful.

Authors reply: We believe that the FeCo alloy with adjustable N-binding energy as well as the suppression of nitridation accounts for its higher activity for ammonia decomposition. Thank you for your interesting suggestion of studying a physical mixture of the Fe/MgO and Co/MgO catalysts in the ammonia decomposition and checking for nitridation. We did the corresponding experiments and the XRD of the reduced mixed catalysts showed clearly the reflections of $\alpha\text{-Fe}$, fcc-Co, and Magnesium wuestite oxide, indicating no FeCo alloy formation during isothermal reduction as expected and in clear contrast to the co-precipitated catalysts (Figure S22 b, SI). After ammonia decomposition, in the XRD of spent $\text{Fe}/\text{MgO-Co}/\text{MgO}$ catalyst mixture, the reflections of $\alpha\text{-Fe}$ disappeared whereas the Fe nitride formed, including Fe_3N and Fe_4N . Meanwhile fcc Co and magnesium wuestite remained. The new iron

nitride Fe_4N formation might indeed be related to the existence of Co/MgO catalyst during reaction, where the reaction atmosphere (NH_3/H_2) was changed. Such effect might thus be strong enough to change the formed iron nitride to a nitrogen-poorer phase, but it did not prevent the nitridation of the Fe phase. Also, the conversion of the physical mixture was in between those of the pure Fe/MgO and Co/MgO. Therefore, the beneficial effect of the $\text{Fe}_{0.5}\text{Co}_{0.5}/\text{MgO}$ catalyst can be assigned to the alloy formation and interpreted as an adjustment of the N-binding energy together with the successful suppression of nitridation, which required the presence of the alloy while the presence of Cobalt in a physical mixture was not enough.

We added this information and discussion in page 13 of the revised manuscript as follows: “To assign the observed suppression of nitridation to the formation of the alloy, a physical mixture of Fe/MgO and Co/MgO was tested in ammonia decomposition. The activity of this mixture was between the activity of pure Fe/MgO and Co/MgO (Figure S22a, SI). After ammonia decomposition, the XRD of the spent catalyst mixture showed the disappearance of the α -Fe reflections while crystalline Fe nitrides formed, including Fe_3N and Fe_4N (Figure S22b, SI). Therefore, the suppression of iron nitridation is assigned to the formation of the alloy and not to the presence of unalloyed cobalt alone.”, and in Figure S22 (page S22) in the Supplementary Information as follows: “

Figure S22 NH_3 conversion of physically mixed Fe/MgO (10 mg) – Co/MgO (10 mg) catalyst in ammonia decomposition in a gas mixture of 3% NH_3/Ar at a flow rate of 80 ml min^{-1} , the NH_3 conversion curves of Fe/MgO and Co/MgO are shown for comparison (a). XRD patterns of reduced physically mixed Fe/MgO – Co/MgO (after isothermal reduction at 600°C) and of the spent mixture (after the ammonia decomposition reaction). (b).

The activity of physically mixed Fe/MgO (10 mg) – Co/MgO (10 mg) is between the activity of Fe/MgO and Co/MgO catalysts. The XRD pattern of the reduced sample shows only the reflections of magnesium wuestite, α -Fe and fcc Co, indicating no FeCo alloy formation. While the XRD pattern of the spent catalyst shows the reflections of iron nitrides (Fe_3N and Fe_4N), magnesium wuestite and fcc Co, indicating clear nitridation of the iron fraction in the mixture. The formation of different nitride species could be related to the presence of active Co/MgO catalyst during the reaction, which changes the local reaction atmosphere towards lower NH_3/H_2 ratio.”

Reviewer #3:

Comments:

This work an MgFeO solid solution, promoted with cobalt as a catalyst for ammonia decomposition. The authors claim that the high activity is associated to the lack of nitridation suppressed by the presence of cobalt, following a well-reported approach consisting of combining metals with weak and strong nitrogen binding energy. In my opinion, this paper contains a number of inconsistencies and thus, I don't think it should be published in Nature Comms.

Authors reply: We thank the reviewer for her / his general comment. Our point-by-point response is given below.

Comment 1): *Promoted iron-based catalysts are active for ammonia decomposition at high temperatures, as the one presented here as reported in a number of recent reviews. The authors do not report low temperature activity and thus, I find the tittle misleading.*

Authors reply: The reviewer's criticism is not easy to understand for us as the reaction temperature is not mentioned in our title. As correctly pointed out by the reviewer, pure iron is only moderately active and required elevated temperature in agreement with our data. We show that by alloying it with cobalt, its activity can be increased and the reaction temperature can be lowered by ?? °C as clearly shown in Figure 4b. For the kinetic evaluation, we have chosen a medium temperature of 500°C for comparison of catalyst with high and low activity to show this effect. We believe that these aspects are represented in the title.

Comment 2): *There is a considerable body of literature following the approach of combining metals with weak and strong nitrogen binding energy to enhance the activity of non-noble metal catalysts, mainly iron and cobalt-based, so this work presents limited novelty. See for example: 10.1016/j.ijhydene.2018.07.085, 10.1016/j.ijhydene.2014.06.081, 10.1016/j.apcatb.2020.119405 in addition to the references in the manuscript 14-18 about iron-based alloys*

Authors reply: Indeed, we mentioned that the concept of combining strong and weak binding elements to nitrogen was introduced, e.g., by Boisen et al. and that we followed this concept. We do not claim that we were the only ones to do that and we thank the reviewer for the additional references regarding this concept, which we have added into the reference list. We would like to clarify that the novelty of our work lies in other important aspects, such as i. the understanding of the observed effects based on DFT calculations, ii. the introduction of iron nitridation in this concept showing the change from strong binding (pure Fe) to weak binding (Fe nitrides), iii. the additional role of cobalt in the suppression of nitridation beyond the moderating effect on the N-binding, and iv. the application of the alloying concept in combination with a precursor-approach to synthesize MgO-supported metal catalysts with unique loading of 74% with high activity and stability. We are thus very confident that this work will not only be of high interest to the academic community, but also to practitioners as it proves the practical potential of a general academic approach and its successful application for ammonia decomposition.

Comment 3): *The authors report TOF numbers based "on the metal particle size distribution assuming fully exposed particle surfaces and equal activity of all surface atoms". If I understand this correctly, they consider that only surface atoms are involved in the reaction and thus, they report considerably TOF numbers that if considering the whole metal amount (which is standard in the literature). If I am correct, I consider this misleading and wrong.*

Authors reply: We disagree with the reviewer's view on the standard utilization of the TOF concept in the literature. To our understanding, the turnover frequency (TOF) was defined in heterogeneous catalysis as the number of reacting molecules per active site per unit time. We are very confident that the vast majority of researchers in heterogeneous catalysis understand catalytic turnover as surface reactions and therefore preferably normalize the reaction rate to surface atoms ("active sites") to obtain TOFs and not to bulk atoms. At least the former definition seems superior to an alternative that is based on the total amount of metal as it accounts for different metal dispersions. Although there are general problems in the counting of active sites, we cannot see how this is misleading or wrong.

Comment 4): The effects on particle sizes presented in Figure S20 should be discussed in the main text as the authors should not discard the effect of size on activity

Authors reply: The reviewer is absolutely correct, we need to deduct the effect / influence of size on the activity. Therefore, we did the evaluation of the TOF with considering the exposed surface atoms (Figure 4b), since different particle size of metal reflects varying metal dispersions (see also our response to comment #3 above). Regarding comments 1-4, no changes were made to the revised manuscript with the exception of the enhanced reference list.

Comment 5): Following previous comment, it is not clear what the catalyst in Fig 1 e STEM figures is – is it after reduction at 600C at 5 h?

Authors reply: Yes, the catalyst used for the Figure 1e STEM EDS map is Fe/MgO after isothermal reduction at 600°C for 5h. According to our nomenclature introduced on page 5, the calcined materials are denoted pre-catalysts, in this case the MgFe₂O₄ spinel, and the term catalyst refers to the reduced sample Fe/MgO. This information was described in the page 5 of the original manuscript by the following part: "for the sake of brevity, we will denote the support phase as MgO and reduced catalysts as Fe/MgO".

Comment 6): The authors should compare their catalytic activity to other iron-based catalysts in the literature as well as Ru-based catalyst, currently being the state-of-the art system for ammonia cracking.

Authors reply: We have compared the activity of our iron catalyst to iron-based catalysts in the literature in Table S5 in the original Supplementary Information. As the reviewer suggested, we have added activity data on Ru-based catalyst in that Table S5 during the revision.

Comment 7): Figure 1d and Figure 3b show the XRD-based identification of species in the (MgFe)O and (MgFeCo)O catalysts showing different species in both catalysts – which one are the active ones?

Authors reply: Figure 1 c, d and Figure 3 a, b show the reduction process of the pre-catalysts MgFe₂O₄ and Mg(Fe_{0.5}Co_{0.5})₂O₄ by XRD. After the isothermal reduction, all catalysts will contain a metallic (mono- / bi- metal) and an oxidic phase (magnesium wuesitite), where the metals or alloy is considered the active phase and the oxide oxide the support phase as discussed on page 5.

Comment 8): I also find confusing that authors talk about Fe/MgO catalyst when they actually have Fe(1-x)Mg(x)O solid solutions. Again this is misleading.

Authors reply: We agree that the Fe/MgO catalyst also contains (MgFe)O solid solution oxide. The reason is that it is hard to fully reduced all transition metal from a MgO-based solid solution. This was not glossed over, but thoroughly explained in the text. Nevertheless, for the sake of brevity and clarity, we denoted support phase as MgO and reduced catalysts as Fe_{1-x}Co_x/MgO. This information was mentioned in the page 5 of the original manuscript as follows: “for the sake of brevity, we will denote the support phase as MgO and reduced catalysts as Fe/MgO”. We cannot see how this is misleading.

Comment 9): The operando SAX experiments in Figure 2 are done under fully reduced conditions which are different to Fig 1d-XRD. If nitride formation happens at 55C – why is low temperature activity low? Do the authors have results for a second run (e.g. after decreasing temperature) to demonstrate that indeed nitride formation is responsible of low activity?

Authors reply: The in-situ XRD experiment of Fig. 1d relates to the catalyst reduction process, not to the working catalyst, i.e. operando conditions as the XAS experiment in Fig. 2. The conditions of catalyst activation were kept similar in both experiments with a holding time at a reduction temperature of 600 °C in diluted hydrogen. Minor differences might arise due to different sample holder systems, such as a quartz capillary reactor used for XAS measurement compared to much larger XRD chamber smaller and the different loading in these reactions (few mg in XAS vs. ca. 20 mg in XRD). We have no reason to believe that these differences have any effect on the conclusions presented in the manuscript.

We mentioned in the original manuscript that all the freshly reduced catalysts were nitridated at 500°C for 5 h in a mixture of 3% NH₃/He prior to the ammonia decomposition measurements (on page 19). Hence, the catalysts, specifically Fe/MgO was ensured to be fully nitridated before the reaction started, which can explain its low activity at low temperatures. This pre-treatment is a prerequisite for obtaining stable catalyst performance and for excluding an overlap of nitridation with catalytic measurement. We also did a second run of reaction for Fe/MgO and its activity is, as expected, exactly the same to the first run (see figure R1 below). It also demonstrated the stability of the catalyst and supports that the formed iron nitride accounts for the low activity of the Fe/MgO catalyst.

Figure R1: Catalytic activity (NH₃ conversion) of Fe/MgO for ammonia decomposition reaction in two runs (the second run of the reaction was directly after the first run of the reaction).

Comment 10): Regarding the experimental method – how do the authors consider the change in the number of moles during the reaction for the calculation of conversion values?

Authors reply: We thank the reviewer for pointing this out. Since we only use 3% NH₃ in He for the ammonia decomposition, the change of volume / moles of gases through ammonia decomposition can be neglected compared to the rest 97% He in the gas mixture. This information has been added in the revised manuscript (page 19) as follows: “The mole and corresponding volume changes of the gas mixture due to reaction and its effect on the conversion data was neglected as it is limited to the only 3% NH₃ in the gas feed and the 97% of diluent He fraction remain unaffected”.

REVIEWERS' COMMENTS

Reviewer #1 (Remarks to the Author):

I went through the revised version of the manuscript, and particularly, the response to the comments and suggestions from the three reviewers. It seems to me the authors have made great efforts to address the raised concerns and to improve the work accordingly. With the supplementary results and the additional interpretations and discussion, the issues are essentially clarified and addressed. On this basis, I would like to recommend the acceptance of this revised version for publication.

Reviewer #2 (Remarks to the Author):

The authors have addressed my comments and concern satisfactorily. I would suggest accepting this work.

Response to reviewer comments on manuscript Nature Communications
(NCOMMS-23-26180A)

“Highly loaded Bimetallic Iron-Cobalt Catalysts for Hydrogen Release from Ammonia”

Reviewer #1:

Comments:

I went through the revised version of the manuscript, and particularly, the response to the comments and suggestions from the three reviewers. It seems to me the authors have made great efforts to address the raised concerns and to improve the work accordingly. With the supplementary results and the additional interpretations and discussion, the issues are essentially clarified and addressed. On this basis, I would like to recommend the acceptance of this revised version for publication.

Authors reply: We very much appreciate the reviewer for the positive view on our work and thank for her / his time and effort to improve the quality of the manuscript.

Reviewer #2:

Comments:

The authors have addressed my comments and concern satisfactorily. I would suggest accepting this work.

Authors reply: We thanks a lot the reviewer for her / his time and effort and the positive view on our work. All the comments have made the manuscript stronger